# Mechanism of completion of peptidyltransferase centre assembly in eukaryotes

Vasileios Kargas[1,2,3†], Pablo Castro-Hartmann[1,2,3†],
Norberto Escudero-Urquijo[1,2,3], Kyle Dent[1,2,3], Christine Hilcenko[1,2,3],
Carolin Sailer[4], Gertrude Zisser[5], Maria J Marques-Carvalho[1,2,3],
Simone Pellegrino[1,2,3], Leszek Wawiórka[1,2,3,6], Stefan MV Freund[7],
Jane L Wagstaff[7], Antonina Andreeva[7], Alexandre Faille[1,2,3], Edwin Chen[8],
Florian Stengel[4], Helmut Bergler[5], Alan John Warren[1,2,3]*

[1]Cambridge Institute for Medical Research, Cambridge, United Kingdom;
[2]Department of Haematology, University of Cambridge, Cambridge, United
Kingdom; [3]Wellcome Trust–Medical Research Council Stem Cell Institute, University
of Cambridge, Cambridge, United Kingdom; [4]Department of Biology, University of
Konstanz, Konstanz, Germany; [5]Institute of Molecular Biosciences, University of
Graz, Graz, Austria; [6]Department of Molecular Biology, Maria Curie-Skłodowska
University, Lublin, Poland; [7]MRC Laboratory of Molecular Biology, Cambridge,
United Kingdom; [8]Faculty of Biological Sciences, University of Leeds, Leeds, United
Kingdom

*For correspondence:
ajw1000@cam.ac.uk

†These authors contributed
equally to this work

Competing interests: The
authors declare that no
competing interests exist.

Reviewing editor: Nahum
Sonenberg, McGill University,
Canada

**Abstract** During their final maturation in the cytoplasm, pre-60S ribosomal particles are
converted to translation-competent large ribosomal subunits. Here, we present the mechanism of
peptidyltransferase centre (PTC) completion that explains how integration of the last ribosomal
proteins is coupled to release of the nuclear export adaptor Nmd3. Single-particle cryo-EM reveals
that eL40 recruitment stabilises helix 89 to form the uL16 binding site. The loading of uL16 unhooks
helix 38 from Nmd3 to adopt its mature conformation. In turn, partial retraction of the L1 stalk is
coupled to a conformational switch in Nmd3 that allows the uL16 P-site loop to fully accommodate
into the PTC where it competes with Nmd3 for an overlapping binding site (base A2971). Our data
reveal how the central functional site of the ribosome is sculpted and suggest how the formation of
translation-competent 60S subunits is disrupted in leukaemia-associated ribosomopathies.
DOI: https://doi.org/10.7554/eLife.44904.001

## Introduction

The assembly of eukaryotic ribosomes involves the concerted action of over 200 *trans*-acting assembly factors. Following their assembly in the nucleus, pre-60S ribosomal subunits are exported to the cytoplasm where they are converted to translation-competent particles. The pre-60S particle attains export competence following release of the GTPase Nog2 by binding of the adaptor protein Nmd3 that recruits the nuclear export receptor Crm1 (Xpo1) through a C-terminal leucine-rich nuclear export signal sequence (*Ho et al., 2000*; *Thomas and Kutay, 2003*; *Trotta et al., 2003*). Crm1-independent adaptors, including Arx1 (*Hung and Johnson, 2006*; *Bradatsch et al., 2007*), Bud20 (*Bassler et al., 2012*), Mex67 (*Yao et al., 2007*) and Ecm1 (*Yao et al., 2010*) facilitate pre-60S nuclear export.

**eLife digest** Biological machines called ribosomes make proteins in the cells of our body. Mammalian cells build roughly 7,500 new ribosomes every minute, each one containing 80 proteins and four RNA molecules. Problems that prevent ribosomes from assembling correctly have been linked to cancers such as leukemia, and a class of disorders called ribosomopathies that increase the likelihood of someone developing cancer. Understanding how ribosomes assemble could therefore help to develop new treatments for these diseases.

Ribosomes are mostly constructed in the cell nucleus, but the final stages of assembly occur in the cytoplasm of the cell. A protein called Nmd3 binds to the partly constructed ribosome to export it out of the nucleus. Then, the final ribosomal proteins integrate into the structure to form a key site called the peptidyltransferase centre (PTC), which is where the ribosome joins together amino acids when making new proteins for the cell. Questions remained about how these final assembly steps occur, and how Nmd3 is removed from the ribosome.

Kargas et al. have now examined how the PTC forms by using a method known as cryo-electron microscopy to determine the structures that the ribosome forms at different stages of assembly. This revealed that when the last two ribosomal proteins integrate into the ribosome, the ribosomal RNA goes through large shape changes that evict Nmd3 from the PTC. Quality control factors then check the structure of the newly formed ribosome and, if it passes their checks that it works correctly, license it to start making cell proteins.

This stage of ribosome assembly is likely to occur in the same way in all plant, animal and other eukaryotic species. The results presented by Kargas et al. will also help researchers to better understand the consequences of the mutations that affect ribosomal proteins in cancer cells. Ultimately, this knowledge may help to uncover new ways to treat cancer and ribosomopathies.

DOI: https://doi.org/10.7554/eLife.44904.002

Once the ribosomal precursor enters the cytoplasm, the final assembly factors are removed and the last remaining ribosomal proteins integrated. The AAA-ATPase Drg1 initiates the final cascade of cytoplasmic events by recycling the assembly factors Rlp24 and Nog1 (*Kappel et al., 2012*; *Pertschy et al., 2007*; *Lo et al., 2010*). Downstream cytoplasmic maturation events include the release and recycling of additional shuttling proteins including the export factors Arx1, Mex67 and Nmd3, removal of the GTPase Lsg1 as well as incorporation of late joining ribosomal proteins. The Arx1-Alb1 heterodimer is bound at the end of the peptide exit tunnel (*Bradatsch et al., 2012*; *Leidig et al., 2014*; *Wu et al., 2016*), from where it is released by the concerted action of the zinc-finger protein Rei1 and the cytosolic J protein Jjj1 (human DNAJC21) that stimulates the ATPase activity of the Hsp70 chaperone protein Ssa (*Hung and Johnson, 2006*; *Lebreton et al., 2006*; *Demoinet et al., 2007*; *Meyer et al., 2007*; *Meyer et al., 2010*; *Lo et al., 2010*).

Incorporation of Rei1 requires prior release of Nog1, whose C-terminal tail seals the exit tunnel as the particle transitions from the nucleolus to the cytoplasm (*Wu et al., 2016*). Interestingly, Rei1 also inserts into the exit tunnel (*Greber et al., 2012*; *Greber et al., 2016*) from where it is displaced during later cytoplasmic maturation steps by Reh1 (*Ma et al., 2017*). However, it is unclear when Reh1 is exchanged for Rei1 and how long Reh1 persists on the particle. A parallel branch of the cytoplasmic maturation pathway involves Yvh1-dependent exchange of uL10 for the placeholder protein Mrt4 to form the P-stalk (*Kemmler et al., 2009*; *Lo et al., 2009*; *Rodríguez-Mateos et al., 2009*).

The subsequent cytoplasmic maturation steps are crucial to correctly shape the peptidyltransferase centre (PTC), evolutionarily the oldest part of the ribosome, in a strictly controlled sequence of events. Single-particle cryo-electron microscopy (cryo-EM) has identified the binding sites for Nmd3 and Lsg1 on the intersubunit face of the 60S subunit (*Ma et al., 2017*; *Malyutin et al., 2017*). Nmd3 spans the tRNA corridor from the uL1 protein at the L1 stalk through the tRNA exit site and the PTC to interact with the anti-association factor eIF6 (yeast Tif6; herein called eIF6) at the sarcin-ricin loop (SRL), while Lsg1 embraces H69. The timing and mechanism of Nmd3 release remains unclear, but is dependent on the GTPase Lsg1 (*Hedges et al., 2005*) and the integration of ribosomal proteins eL40 (*Fernández-Pevida et al., 2012*) and uL16 (*Hedges et al., 2005*).

The eIF6 protein prevents premature joining of the ribosomal 60S and 40S subunits by binding to the SRL and ribosomal proteins uL14 and eL24 (*Gartmann et al., 2010*). Release of eIF6 in the cytoplasm is catalysed by EFL1 (elongation factor like GTPase 1, an EF-2 homolog) and its cofactor SBDS (Shwachman-Bodian-Diamond syndrome, yeast Sdo1) (*Bécam et al., 2001*; *Senger et al., 2001*; *Menne et al., 2007*; *Finch et al., 2011*; *Wong et al., 2011*; *Weis et al., 2015*). The recruitment of SBDS to the 60S subunit depends on the prior integration of uL16 into the PTC in vivo (*Weis et al., 2015*). Although recent structural studies suggest that eIF6 is released after Nmd3 (*Weis et al., 2015*; *Ma et al., 2017*), an alternate model posits that the binding of uL16 breaks the interaction of the Nmd3 N-terminus with eIF6, allowing the recruitment of SBDS to promote eIF6 removal prior to the eviction of Nmd3 (*Lo et al., 2010*; *Malyutin et al., 2017*; *Patchett et al., 2017*).

Maintaining the fidelity of late cytoplasmic 60S subunit maturation is crucial for all eukaryotic cells as defects in this process cause human developmental defects and cancer predisposition. Multiple mutations associated with human bone marrow failure and leukaemia target this pathway. For example, recurrent somatic mutations in the *RPL10* gene (encoding the ribosomal protein uL16) have been identified in 10% of cases of paediatric T-cell acute lymphoblastic leukaemia (T-ALL) (*De Keersmaecker et al., 2013*). The T-ALL associated uL16-R98S missense variant impairs the release of both Nmd3 and eIF6 (*De Keersmaecker et al., 2013*). However, the underlying mechanism is unknown. Inherited mutations in the *SBDS* gene have been identified in 90% of individuals with Shwachman-Diamond syndrome (SDS), an autosomal recessive disorder characterized by poor growth, exocrine pancreatic insufficiency, skeletal abnormalities and bone marrow failure with an increased risk of progression to myelodysplastic syndrome (MDS) and acute myeloid leukaemia (AML) (*Boocock et al., 2003*; *Warren, 2018*). SDS is also associated with mutations in *DNAJC21*, the human homologue of yeast *JJJ1* (*Dhanraj et al., 2017*; *Tummala et al., 2016*; *D'Amours et al., 2018*) and *EFL1* (*Stepensky et al., 2017*).

To elucidate the mechanism of PTC completion and understand how this process in corrupted by leukaemia-associated mutations, we used single-particle cryo-EM and cross-linking mass spectrometry to delineate the sequential steps that lead to assembly of the key functional site of the ribosome. We show how integration of the final ribosomal proteins eL40 and uL16 initiates a hierarchical sequence of RNA and protein rearrangements that result in release of the essential nuclear export adaptor Nmd3, a key conserved step in PTC formation. Defective completion of the PTC causes developmental disorders associated with an increased propensity for malignant transformation. Hence, our atomic models not only illuminate the mechanism of PTC assembly but also suggest how mutations found in leukaemia disrupt this process.

## Results

### Overview of the pre-60S Lsg1 particles

We set out to determine the mechanism of cytoplasmic 60S subunit maturation and completion of PTC assembly by affinity purifying native pre-60S particles from *S. cerevisiae* using tandem affinity purification (TAP)-tagged Lsg1 as bait (*Figure 1—figure supplement 1A*) and subjecting them to immunoblotting (*Figure 1—figure supplement 1B*), cryo-EM analysis (*Figure 1A*, *Figure 1—figure supplement 1C–F*, *Figure 1—figure supplements 2–4*, *Supplementary file 1A, B*) and crosslinking mass spectrometry (XL-MX) (*Figure 1—figure supplement 5A,B*, *Supplementary file 2*). Immunoblotting revealed enrichment of the assembly factors Lsg1, Nmd3, Arx1 and eIF6 in the purified particles, but under-representation of ribosomal proteins uL16 and uL10 (P0) (*Figure 1—figure supplement 1B*). Analysis by single-particle cryo-EM with extensive 3D classification yielded a series of structures (hereafter termed states I-VI) that likely reflect sequential snapshots of final cytoplasmic pre-60S maturation (*Figure 1A*). The ability to capture state VI (lacking both Nmd3 and Lsg1) likely reflects ongoing maturation of the particles or possibly the dissociation of Lsg1 and Nmd3 during immunopurification. We refined all six pre-60S cryo-EM reconstructions to average resolutions of 3.1–3.9 Å with the local resolution extending to 2.3 Å in the core of the particles in states I and III (*Figure 1—figure supplement 1E,F*). The maps allowed us to fit and refine atomic models for all six assembly states, including the biogenesis factors Arx1, Lsg1, Nmd3, Rei1, Reh1, eIF6, ribosomal proteins eL40, uL16 and uL11 together with the 5S, 5.8S and 25S rRNA (*Figure 1B*, *Figure 1—figure supplements 2–4*, *Supplementary file 1A, B*). In addition, we identified density corresponding to

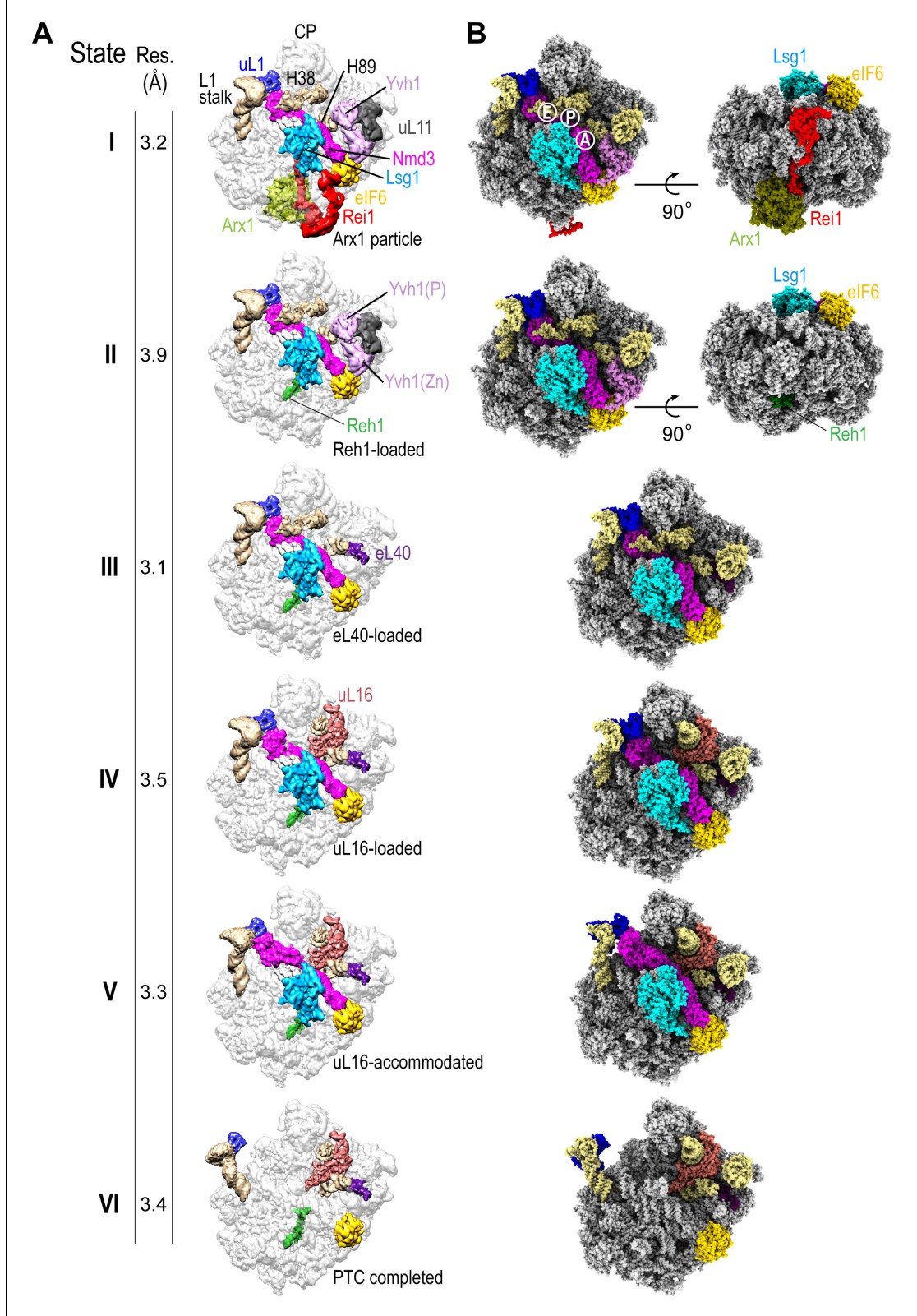

**Figure 1.** Sequential steps in late cytoplasmic 60S subunit maturation. (**A**) Cryo-EM reconstructions of six cytoplasmic maturation states (I–VI). States, overall resolution, changes in protein composition and rRNA conformation are indicated. The phosphatase (P) and zinc finger (Zn) domains of Yvh1 are indicated in state II. (**B**) Atomic models of pre-60S states I-VI with rRNA and biogenesis factors highlighted.

DOI: https://doi.org/10.7554/eLife.44904.003

*Figure 1 continued on next page*

*Figure 1 continued*

The following figure supplements are available for figure 1:

**Figure supplement 1.** Sample purification, cryo-EM image analysis and local resolution of pre-60S particles.
DOI: https://doi.org/10.7554/eLife.44904.004

**Figure supplement 2.** Cryo-EM data processing scheme.
DOI: https://doi.org/10.7554/eLife.44904.005

**Figure supplement 3.** Cross-validation against overfitting.
DOI: https://doi.org/10.7554/eLife.44904.006

**Figure supplement 4.** Local resolution for ribosome assembly factors, eL40 and uL16.
DOI: https://doi.org/10.7554/eLife.44904.007

**Figure supplement 5.** Protein-protein interaction network of pre-ribosomal particles purified by Lsg1-TAP.
DOI: https://doi.org/10.7554/eLife.44904.008

the phosphatase (*Baßler et al., 2017*) and zinc finger (*Zhou et al., 2019*) domains of Yvh1 situated between uL11 in the P-stalk and eIF6 (*Figure 1A*, states I-II).

Sequential maturation of the cytoplasmic pre-60S subunit couples dramatic conformational rearrangements of two long flexible helices (the L1 stalk and H38) to recruitment of the final ribosomal proteins (eL40 and uL16) and assembly factor displacement (*Figure 1A*). In states I-III, the L1 stalk is displaced inwards with H38 in the 'closed' position. Rearrangement of H38 to the mature position (state IV) is followed by partial (state V) then full (state VI) retraction of the L1 stalk. Focused classification of the local density around the P-stalk (*Figure 1—figure supplement 2*) revealed that binding of ribosomal protein eL40 does not occur until state III, prior to the recruitment of uL16 in state IV. Although there is clear density for uL11 (state I) at the base of the P-stalk, uL10 was poorly defined in our maps, consistent with the paucity of uL10 protein in the Lsg1-TAP particles as detected by immunoblotting (*Figure 1—figure supplement 1B*). Focused classification around the P-stalk revealed that Yvh1 is predominantly bound in state I, is present at low occupancy in state II but is absent in state III. Thus, Yvh1 and eL40 do not appear to bind simultaneously to the same particle.

The homologous C-termini of Reh1 and Rei1 occupy the PET sequentially (*Figure 1A*). While the departure of Rei1 seems to be concurrent with Arx1 release, the C-terminus of Reh1 unexpectedly persists in the exit tunnel even after the departure of Nmd3 and Lsg1. The timing of the exchange of Rei1 for Reh1 (between states I-II) is consistent with XL-MS analysis that yielded crosslinks between Rei1, Arx1 and eL24, while Reh1 yielded crosslinks to eL24, but not to Rei1 or Arx1 (*Figure 1—figure supplement 5A,B* and *Supplementary file 2*). These data are also consistent with a previous structural snapshot (*Ma et al., 2017*) and with co-immunoprecipitation analysis (*Parnell and Bass, 2009*). We conclude that the exchange of Reh1 for Rei1 occurs at the time of Arx1 release, but that surveillance of the PET by Reh1 continues throughout the entire process of cytoplasmic pre-60S maturation.

While the overall structures of Rei1 and Reh1 are consistent with previous reports (*Greber et al., 2016*; *Ma et al., 2017*), our maps reveal additional density that corresponds to the N-termini of both Rei1 and Reh1 extending up across the surface of eL24 to directly interact with eIF6 subdomains C and D and with a loop that extends out from the α-helical domain of Lsg1 (residues G438-T456) (*Figure 1A*, state I). The Lsg1 α-helical domain (residues G474-D479) also interacts with uL14.

We unambiguously distinguished the helical C-termini of Reh1 and Rei1 within the PET based on specific side chain densities (*Ma et al., 2017*). Interestingly, the position of the extreme C-terminal leucine and glutamine residues of Rei1 and Reh1 differs compared with previous reports (*Ma et al., 2017*; *Greber et al., 2016*): the side chains of Q393 (Rei1) and Q432 (Reh1) form an electrostatic interaction with the base of U2875 (H89), while the backbone interacts with the base of U2978 (H93).

The coexistence of eIF6 and Reh1 on the same particle (*Figure 1B*) suggests that state VI is not simply a product of eIF6 rebinding but is a bona fide late pre-60S subunit maturation intermediate that lies downstream of Nmd3 and Lsg1 release. These data support the hypothesis that eIF6 is evicted after Nmd3 (*Weis et al., 2015*).

## Nmd3 is assembled from existing ribosomal machinery

We set out to understand the mechanism of Nmd3 release, a key event in the completion of PTC assembly. Nmd3 extends across the entire tRNA binding cleft on the intersubunit face of the 60S subunit from uL1 at the L1 stalk through the E, P and A sites to contact Lsg1 and eIF6 at the SRL (*Figure 1A,B*). The cryo-EM density allowed us to build a complete atomic model for Nmd3 (residues T16-R404) including backbone atoms and side chains (*Figure 2A*), revealing a multi-domain architecture that includes two treble clef zinc fingers (that superimpose with an RMSD of 1 Å over 13 Cα atoms; *Figure 2A*, inset), two alpha-beta domains and two beta-barrel domains. The N-terminal treble clef zinc finger (residues 16–42, C19-C22-C35-C38) is grafted into an alpha-beta domain (residues 43–154) that is structurally related to ribosomal protein eL31. The second alpha-beta domain (residues 155–250) is structurally related to ribosomal protein eL22, while the two C-terminal beta-barrel domains have SH3 (251–310) and OB (311-400) folds, respectively. Although the combination of the SH3 and OB domains is similar in sequence and structure to eIF-5A-1, the domains are oriented differently with respect to each other. Thus, Nmd3 comprises a modular assembly of existing structural blocks associated with the ribosomal machinery combined together to form a functionally distinct protein.

## Nmd3 promotes H38 closure by stabilising a base flip at A2971

We sought to understand how Nmd3 maintains the 'closed' orientation of the L1 stalk (states I-IV) and H38 (states I-III) (*Figure 1A*). The Nmd3 OB domain holds the L1 stalk in the closed position, while the eL22-like, SH3 and OB domains, together with an extended C-terminal loop, encircle and distort the tip of H38 (*Figure 2B,C*). Specifically, two loops (β1–2 and β3–4) from the OB domain stabilize a base-flip of A1025. The side chain and backbone atoms of N332 contact the base and backbone phosphate of A1025. H364 contacts the backbone phosphate of A1027, while the side chains of Y402 and R404 contact the backbone phosphates of A1026-G1029. Within the SH3 domain, the side chain of K253 contacts the phosphate backbone of C1023. Helix α1 from the eL22-like domain (particularly the side chains of R169 and Q176) packs against the bases and backbone of H38 (G1020-C1023).

## Nmd3 contacts Lsg1 and eIF6 throughout cytoplasmic maturation

Throughout the steps of pre-60S subunit maturation visualised herein, H69 adopts a conformation that differs from the mature ribosome (*Ben-Shem et al., 2011*) but that surprisingly persists even in the absence of Nmd3 and Lsg1 (state VI) (*Figure 3A,B*). Lsg1 stabilises a base flip at G2261 (H69) (*Figure 3A*), while the side chain of Lsg1 W142 stacks against the base of A2256 at the tip of H69 (*Figure 3B*). The altered conformation of H69 is also promoted by a β-hairpin in the SH3 domain of Nmd3 that stabilises a base flip in U2269 at the junction between H68 and H69 (*Figure 3C*). The Nmd3 SH3 domain makes additional interactions with the 25S rRNA using two short α-helices (SH3-α1 and SH3-α2) (*Figure 3D*). As the altered conformation of H69 is maintained even in the absence of both Nmd3 and Lsg1 (state VI), we suggest that H69 may only adopt the mature conformation after joining of the 60S and 40S subunits.

Compared with the reconstituted Nmd3-Lsg1-60S particle (*Malyutin et al., 2017*), the Nmd3 eL31-like domain is rotated (60°) away from H89 towards the long α-helix of the Lsg1 GTPase domain in each of the native intermediates carrying Lsg1 and Nmd3 (states I-V) (*Figure 1A*). The interaction involves a stacking interaction between the side chain of Nmd3 residue W104 and the side chain of Lsg1 residue R163 (*Figure 3E*). Although focused classification of state III identified a subset of particles in which the Nmd3 eL31-like domain is rotated towards H89, this class lacks Lsg1 and Arx1, with Reh1 present in the PET. We suggest that the rotated conformation of the Nmd3 eL31-like domain in this subset of state III particles is a consequence of Lsg1 dissociation during sample preparation rather than a physiologically relevant 'pre-Lsg1' state.

We next assessed the dynamic properties of the N-terminus of Nmd3 in solution using NMR spectroscopy. Heteronuclear ¹H {¹⁵N} NOE analysis of the *Archaeoglobus fulgidus* Nmd3 (residues 22–150) indicates that the N-terminal zinc finger (residues 22–43) has a higher degree of motion than the attached eL31 domain on the picosecond timescale (*Figure 3—figure supplement 1A,B*). Exchange broadened residues indicate the presence of a flexible linker between the two domains that is undergoing segmental motion. These data indicate that the Nmd3 N-terminal zinc finger has

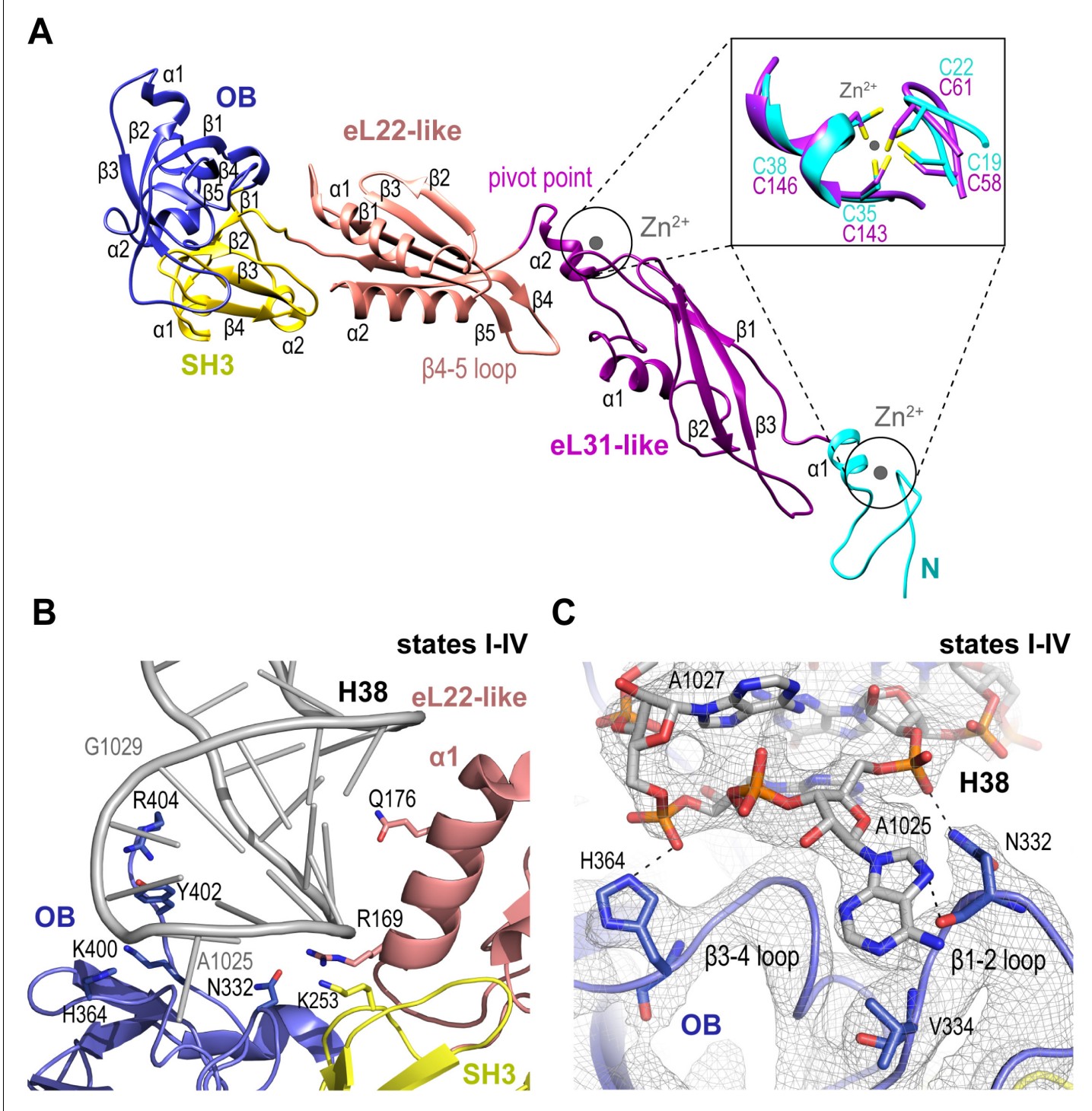

**Figure 2.** Nmd3 maintains the closed conformation of helix 38. (**A**) The modular atomic structure of Nmd3 with the individual domains (OB, SH3, eL22-like, eL31-like and the two treble clef zinc fingers) indicated by colour coding. Inset shows the superposition of the two treble clef zinc fingers (RMSD ~1 Å). (**B**) Interaction of the Nmd3 OB, SH3 and eL22-like domains with the tip of H38. (**C**) Nmd3 stabilises an A1025 base-flip. Atomic model, derived from the state III map (grey mesh), shows the interaction between Nmd3 and the flipped-out base A1025.

DOI: https://doi.org/10.7554/eLife.44904.009

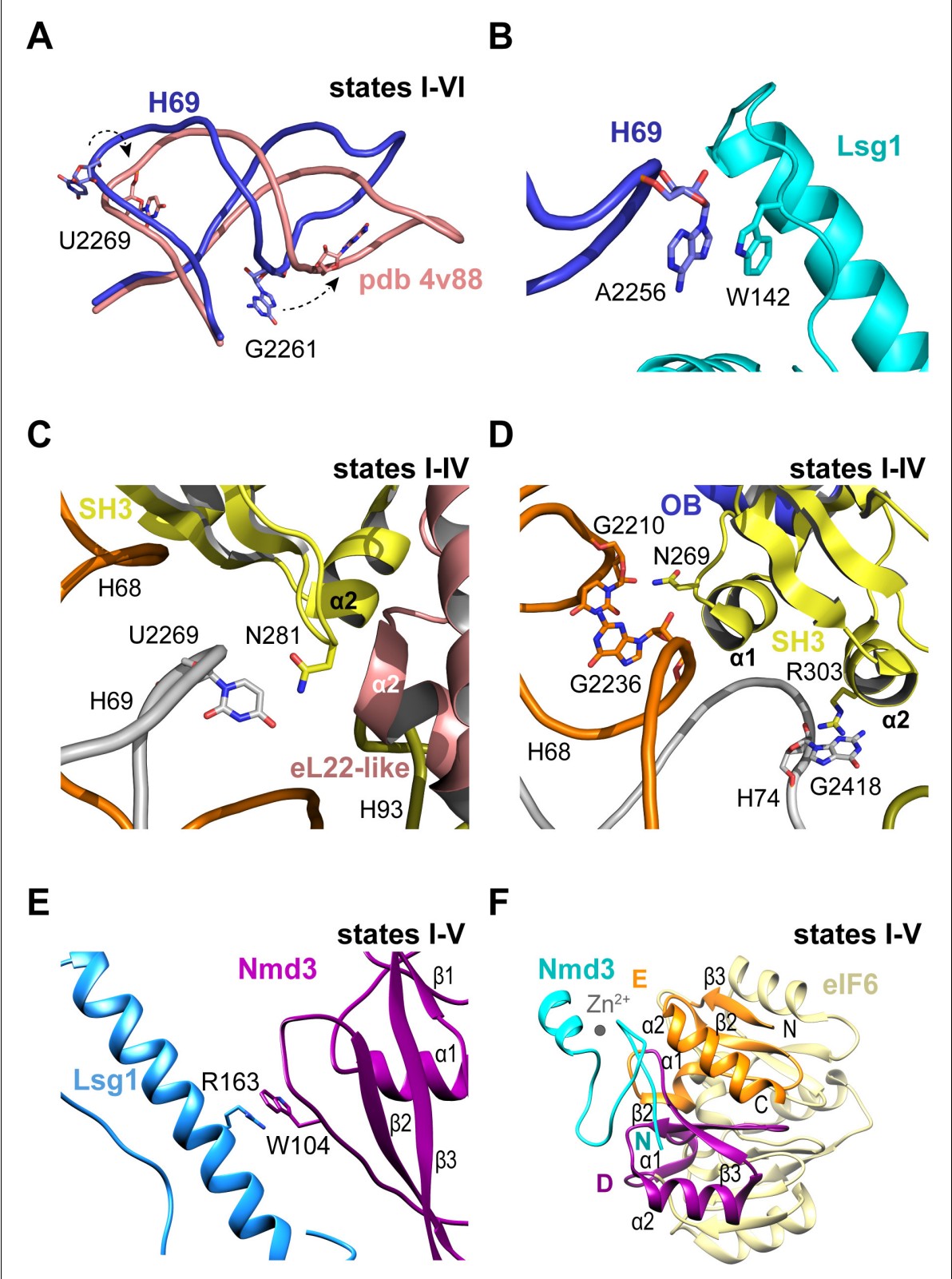

**Figure 3.** Nmd3 interacts with Lsg1 and eIF6 throughout 60S maturation. (**A**) Conformation of H69 (states I-VI) differs from the mature 60S subunit (4v88, *Ben-Shem et al., 2011*). (**B**) Interaction of Lsg1 with the tip of H69. (**C**) Contact between Nmd3 residue N281 and base U2269 (H69). (**D**) Interactions of Nmd3 SH3 domain with H68 and H74. (**E**) Interaction between Lsg1 and the Nmd3 eL31-like domain. (**F**) Interaction of the Nmd3 N-terminal domain with the D and E domains of eIF6.

*Figure 3 continued on next page*

*Figure 3 continued*

DOI: https://doi.org/10.7554/eLife.44904.010

The following figure supplement is available for figure 3:

**Figure supplement 1.** NMR spectroscopic analysis of *A.fulgidus* Nmd3.

DOI: https://doi.org/10.7554/eLife.44904.011

dynamic mobility that allows it to sample a range of positions in solution. However, the cryo-EM data indicate that the Nmd3 zinc finger remains bound to eIF6 subdomains D and E (total buried surface area of 107 Å$^2$) throughout states I-V (*Figure 3F*), providing a click-lock that fixes the Nmd3 N-terminus in position throughout pre-60S maturation. Taken together, our data illustrate how the modular architecture of Nmd3 confers it with the flexibility to modulate its conformation depending on the specific stage of pre-60S maturation.

## Loading of eL40 stabilises H89 to facilitate uL16 recruitment

In the late nuclear pre-60S particle purified using epitope-tagged Nog2, the N-terminal domain of Nog1 separates H89 into two strands, H89 adopting an upright position (*Wu et al., 2016*). We therefore sought to understand how H89 accommodates into its mature conformation to form one face of the u16-binding site. Focused classification around the H89 density revealed that in the absence of eL40 (states I-II), H89 adopts a range of conformations even with Yvh1 present (*Figure 4A*). However, following the departure of Yvh1, eL40 stabilises H89 in its near-mature conformation by forming two major contacts with the opposing H91, including a stacking interaction of A2847 (H89) with G2898 (H91) and base pairing between C2844 (H89) and G2898 (H91) (*Figure 4B*). An observed contact between Yvh1 and the side chain of uL6 K141 overlaps with the binding site for the N-terminus of eL40 in state III, raising the possibility that the recruitment of eL40 may destabilise Yvh1 to promote its departure from the pre-60S particle. In turn, the docking of uL16 to the upper surface of H89 promotes a dramatic ~65° rotation of H38 away from Nmd3 (*Figure 4C*), sandwiching uL16 in a cleft between H38 and H89 (*Figure 4D*).

## A conformational switch in Nmd3 drives PTC completion

In mature, actively translating ribosomes with tRNA bound, the eukaryotic-specific loop (residues 102–112) of uL16 extends into the PTC (*Schmidt et al., 2016*). However, in states I-IV the eL22-like domain of Nmd3 extends into the PTC where it is surrounded by helices H64, H69-71, H80, H90 and H92-93 (*Figure 5A*). Residues on the surface of the eL22-like domain β-sheet make extensive interactions with the backbone of H80 (*Figure 5B*). Residue N205 (α2 helix) stabilises a stacking base pair interaction between C2308 (H64) and C2284 (H70) (*Figure 5C*). Furthermore, the β4–5 loop of Nmd3 stabilises the 'closed' conformation of base A2971 through specific interactions with the side chains of K204 (helix α2), K224, F242 (strand β5), Y240 and the backbone amide of S238 (β4–5 loop) (*Figure 5D*). As a result, the uL16 P-site loop is unable to extend into the PTC in state IV because of a steric clash with the Nmd3 β4–5 hairpin loop.

The initial docking of uL16 is fully compatible with retention of Nmd3 on the pre-60S particle (*Figure 1A*, *Figure 6A,B*). However, uL16 liberates H38 from the C-terminus of Nmd3 promoting partial (20°) (state V) and subsequently full (56°) retraction (state VI) of the L1 stalk (*Figure 6B,C*). Partial retraction of the L1 stalk (state V) is accompanied by a conformational switch in Nmd3 in which the OB, SH3 and eL22-like domains are displaced upwards and outwards from the P and E sites by ~20 Å through a pivot point between the flexible helical linker connecting the eL22- and eL31-like domains, while the N-terminus remains anchored to eIF6 and Lsg1 (*Figure 6B,D*). Displacement of the eL22-like domain breaks the interaction of residue N205 (helix α2) with C2308 (H64) and C2284 (H70) (*Figure 5C*), allowing helix α2 to form a new contact with H69 (*Figure 6—figure supplement 1*). Displacement of the β4–5 hairpin loop from the entrance to the PTC (state V) permits the uL16 P-site loop to extend into the PTC where the highly conserved, essential residue R110 coordinates A2971 in an 'open' flipped-up conformation (*Figure 6B,C*).

Genetic and biochemical analysis are consistent with an essential role for the uL16 P-site loop in Nmd3 release (*Hofer et al., 2007*; *Bussiere et al., 2012*; *Sulima et al., 2014*; *Weis et al., 2015*; *Patchett et al., 2017*). Furthermore, an *nmd3-N205D* allele rescued the fitness defect of yeast cells

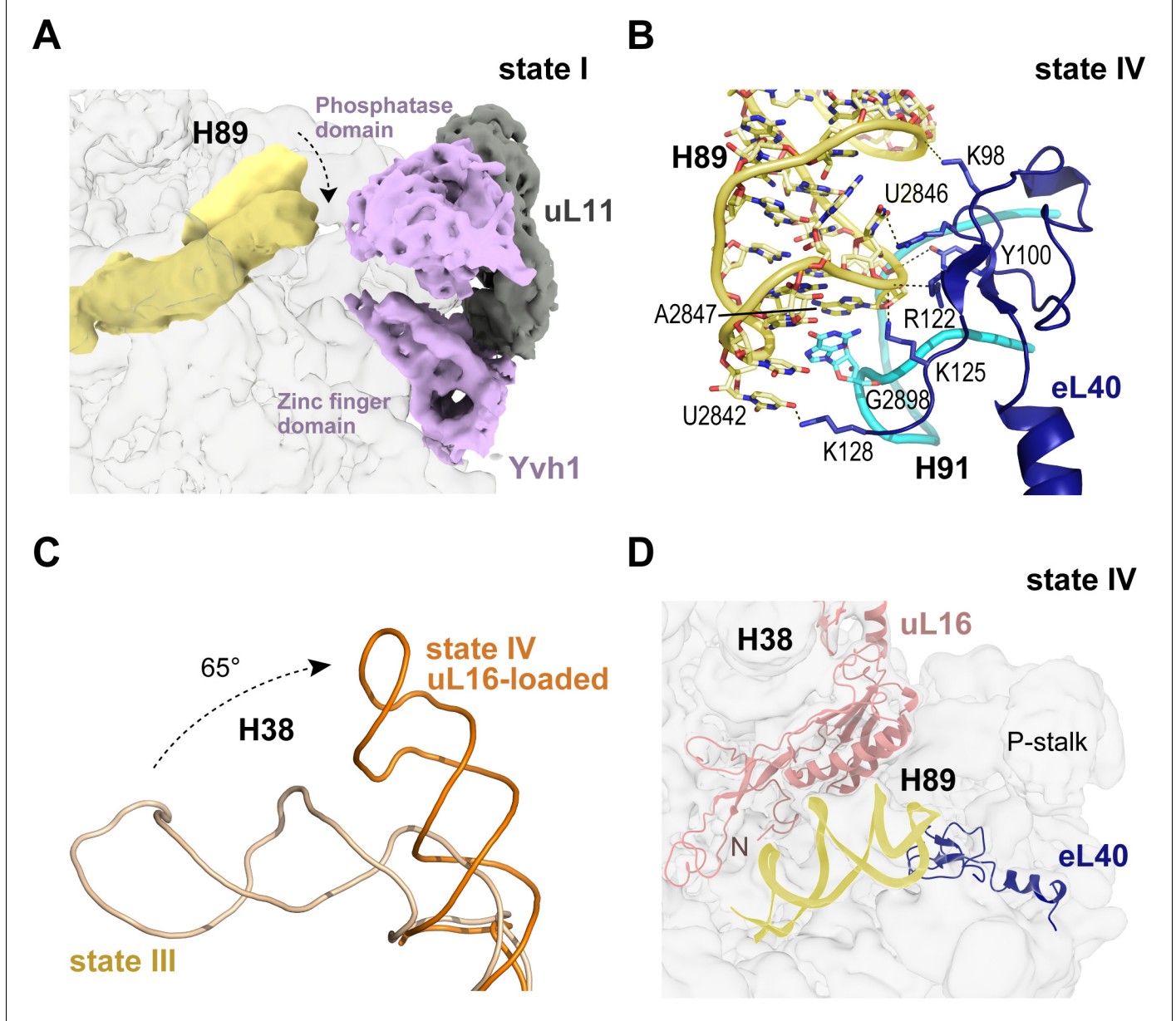

**Figure 4.** Binding of eL40 stabilises H89 to form the uL16 binding site. (**A**) H89 is flexible in states I-II. Cryo-EM density map of Yvh1 is shown in purple; uL11 in grey. The mature conformation of H89 is shown in ribbon representation for comparison. (**B**) Atomic model of the interaction between eL40 and the tip of H89 and H91 (state IV). (**C**) Loading of uL16 induces a rotation of H38. The conformations of H38 in state III (wheat) compared with state IV (orange). (**D**) Conformation of H89 with eL40 and uL16 bound (state IV).
DOI: https://doi.org/10.7554/eLife.44904.012

expressing the T-ALL-associated uL16-R98S mutant as the sole copy of uL16 (*Figure 6—figure supplement 4*). These data support a functional role for the interaction between Nmd3 residue N205 and helices H64 and H70 (*Figure 5C*) in stabilising the binding of Nmd3 to the 60S subunit. We conclude that the uL16 P-site loop drives the conformational equilibrium in favour of Nmd3 release by competing with Nmd3 for an overlapping binding site within the PTC.

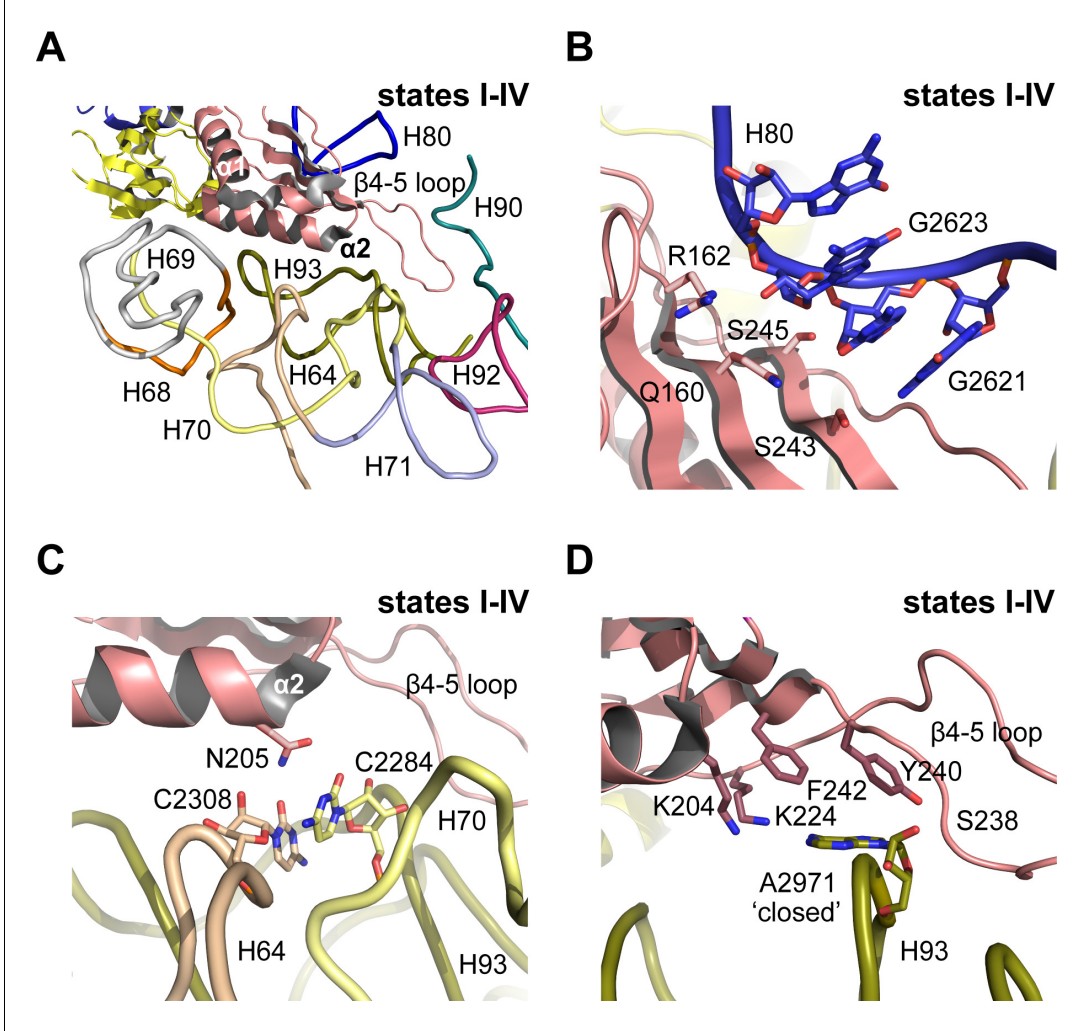

**Figure 5.** Nmd3 masks base A2971 in the peptidyltransferase centre. (**A**) Interactions of Nmd3 with the PTC. (**B**) Interactions of the Nmd3 eL22-like domain β-sheet with H80. (**C**) Nmd3 residue N205 (helix α2) stabilises the junction between bases C2308 (H64) and C2284 (H70). (**D**) The β4–5 loop of the Nmd3 eL22-like domain maintains base A2971 (H93) in the 'closed' conformation.

DOI: https://doi.org/10.7554/eLife.44904.013

## Discussion

### Mechanism of completion of PTC assembly

The cryo-EM structures reported here allow us to propose a mechanism for the completion of assembly of a functional PTC (*Figure 7*, *Video 1*). The sequential incorporation of eL40 and uL16 couples large-scale rRNA rearrangements to a conformational switch in Nmd3 that allows the P-site loop of uL16 to fully accommodate into the PTC to push the conformational equilibrium towards Nmd3 dissociation (*Figure 6A–D*, *Figure 6—figure supplement 1*). The incorporation of eL40 (state III) stabilises H89 to form one face of the uL16-binding platform and may also facilitate Yvh1 release by disrupting its interaction with uL6. While the initial docking of uL16 to the upper surface of H89 is still compatible with the retention of Nmd3 on the particle, it promotes a ~ 65° rotation of H38 away from Nmd3 to adopt its mature conformation, thereby sandwiching uL16 in the cleft formed by H38 and H89. Loss of the stabilising interactions between Nmd3 and the tip of H38 promotes partial retraction of the L1 stalk (state V). This is coupled to a conformational switch in the C-terminus of Nmd3 that displaces the SH3 and OB domains from the E site and the eL22-like domain from the P-site. The uL16 P-site loop now fully accommodates into the PTC by competing with Nmd3 for an

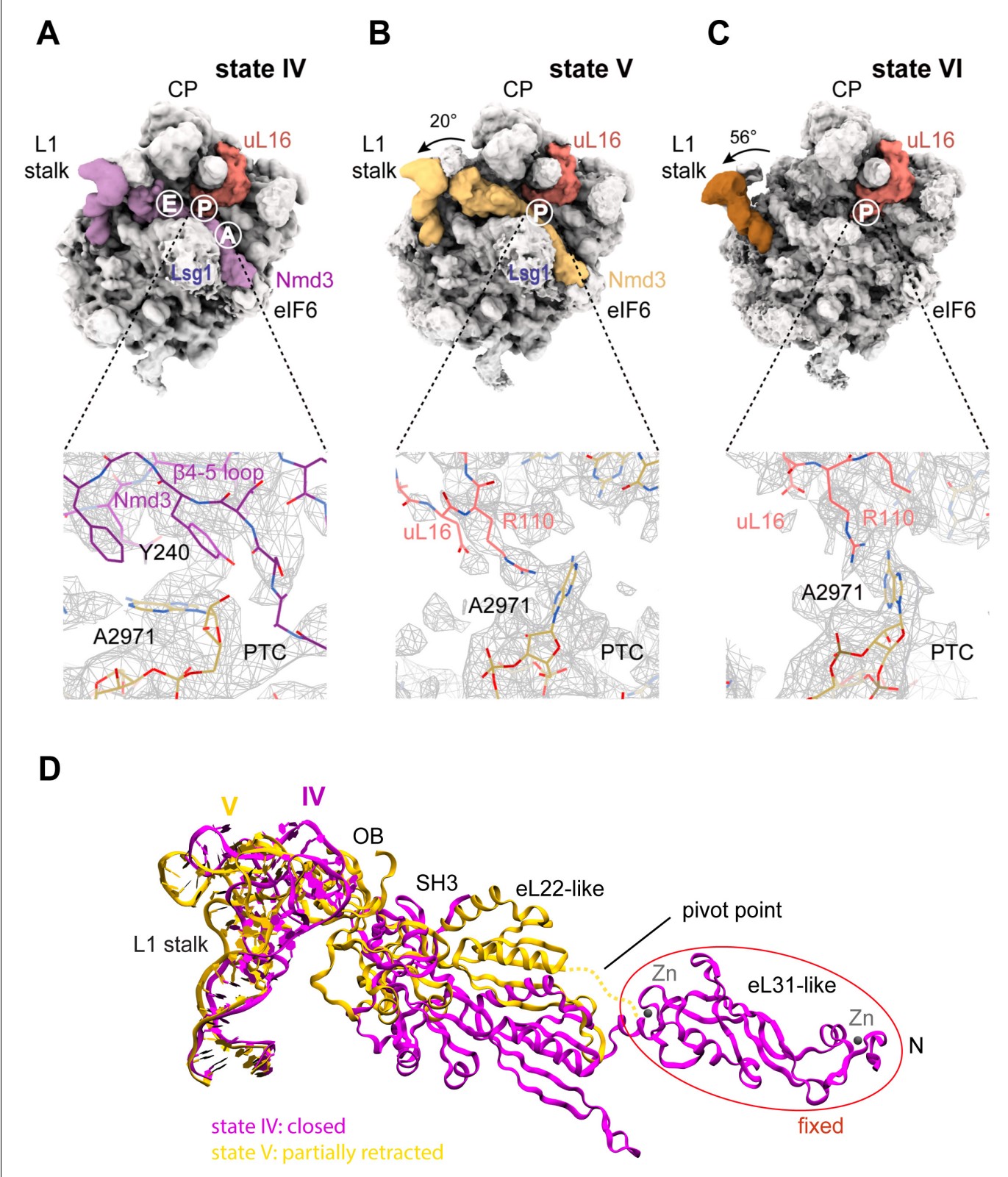

**Figure 6.** Mechanism of Nmd3 release. (**A**) The Nmd3 β4–5 loop masks base A2971 (state IV). Insert shows base A2971 in the 'closed' position. H38 is 'open', the L1 stalk is 'closed' and uL16 is loaded. The A, P and E tRNA binding sites are indicated. (**B**) The uL16 P-site loop competes with the Nmd3 β4–5 loop for binding to base A2971 (state V). The L1 stalk is partially retracted (rotated by 20° relative to the fully closed L1 stalk); the Nmd3 β4–5 loop is displaced from the PTC. The uL16 P-site loop is bound to base A2971 in the flipped-out conformation. (**C**) The completed PTC (state VI). The L1 stalk

*Figure 6 continued on next page*

*Figure 6 continued*

is fully open (rotated by 56°), Nmd3 and Lsg1 have dissociated and the uL16 P-site loop is bound to base A2971 in the flipped-out conformation. (D) L1 stalk retraction promotes a conformational switch in the Nmd3 eL22-like domain. Atomic models for the L1 stalk and Nmd3 in states IV (magenta) and V (gold) are superimposed. The rotational pivot point between the eL31- and eL22-like domains is indicated.

DOI: https://doi.org/10.7554/eLife.44904.014

The following figure supplements are available for figure 6:

**Figure supplement 1.** Release of Nmd3 eL22-like domain involves new contacts with H69.

DOI: https://doi.org/10.7554/eLife.44904.015

**Figure supplement 2.** Structural conflicts during the completion of PTC assembly.

DOI: https://doi.org/10.7554/eLife.44904.016

**Figure supplement 3.** Quality control assessment of PTC assembly by SBDS.

DOI: https://doi.org/10.7554/eLife.44904.017

**Figure supplement 4.** Suppression of the T-ALL associated uL16-R98S variant by mutations in *NMD3*.

DOI: https://doi.org/10.7554/eLife.44904.018

overlapping binding site at base A2971, thus driving the conformational equilibrium in favour of Nmd3 dissociation. The side chain of uL16 residue R110 stabilises the flipped-out conformation of base A2971 to complete the assembly of a functional PTC.

Our structures explain the essential roles of eL40 (*Fernández-Pevida et al., 2012*) and the u16 P-site loop (*Patchett et al., 2017*; *De Keersmaecker et al., 2013*; *Hofer et al., 2007*) in Nmd3 release. The extension of the uL16 P-site loop into the PTC that is observed in the mature ribosome in the presence of tRNA (*Schmidt et al., 2016*) is accomplished by a conformational switch in Nmd3 that allows uL16 to bypass the steric clash with the β4–5 loop of the Nmd3 eL22-like domain (*Figure 6—figure supplement 2A*). Importantly, the link between the Nmd3 N-terminus and eIF6 remains unbroken at this stage of the maturation process, contrary to a previous model (*Malyutin et al., 2017*). Indeed, the links between Nmd3, Lsg1 and eIF6 are maintained throughout states I-V.

Consistent with the studies on nucleolar pre-60S subunit maturation (*Kater et al., 2017*; *Sanghai et al., 2018*), our work does not support parallel 60S assembly pathways as suggested for the bacterial 50S subunit (*Davis et al., 2016*). In contrast, our data strongly support the concept that eukaryotic cytoplasmic 60S maturation proceeds in a linear, stepwise manner.

## Mechanism of activation of Lsg1 GTP hydrolysis

The precise role of Lsg1 in Nmd3 release and the mechanism of Lsg1 GTPase activation remain unclear. It has been proposed that coupling of the flipped-out base G2261 in H69 to the Lsg1 Switch one region (that coordinates a magnesium ion in the active site) may regulate Lsg1 GTP hydrolysis (*Malyutin et al., 2017*). However, the G2261 base-flip appears to persist throughout all the stages of cytoplasmic 60S maturation visualised, even in the absence of Lsg1 (*Figure 3A*). By contrast, the Nmd3 eL22-like domain establishes a new contact with H69 as the L1 stalk retracts (*Figure 6D*, *Figure 6—figure supplement 1*). We speculate that this interaction of Nmd3 with H69 may transiently reposition base G2261, thereby relaying the change in conformation to the Switch one region to activate Lsg1 GTP hydrolysis.

The identification of a crosslink between the N-terminus of Lsg1 and the P-site loop of uL16 (*Figure 1—figure supplement 5A,B* and *Supplementary file 2*) raises the intriguing possibility that Lsg1 may proofread accommodation of the uL16 P-site loop into the PTC as the Nmd3 β4–5 loop is retracted. In this model, Lsg1 may act as a gatekeeper that licenses further progression of 60S subunit maturation by dissociating only once it senses that uL16 is correctly integrated. The disengagement of Lsg1 from the N-terminus of Nmd3 will further destabilise the interaction of Nmd3 with the 60S subunit, allowing retraction of the L1 stalk to pull Nmd3 completely off the intersubunit face.

## Eviction of Nmd3 precedes the release of eIF6

Unexpectedly, our data reveal that the C-terminus of Reh1 is retained in the PET in a late assembly intermediate carrying eIF6 that is downstream of Lsg1 and Nmd3 release (state VI) (*Figure 1A*). Identification of this particle provides compelling evidence that eIF6 release by SBDS and EFL1 occurs after the dissociation of Nmd3 and Lsg1 as proposed (*Weis et al., 2015*) (*Figure 7*). This timing of

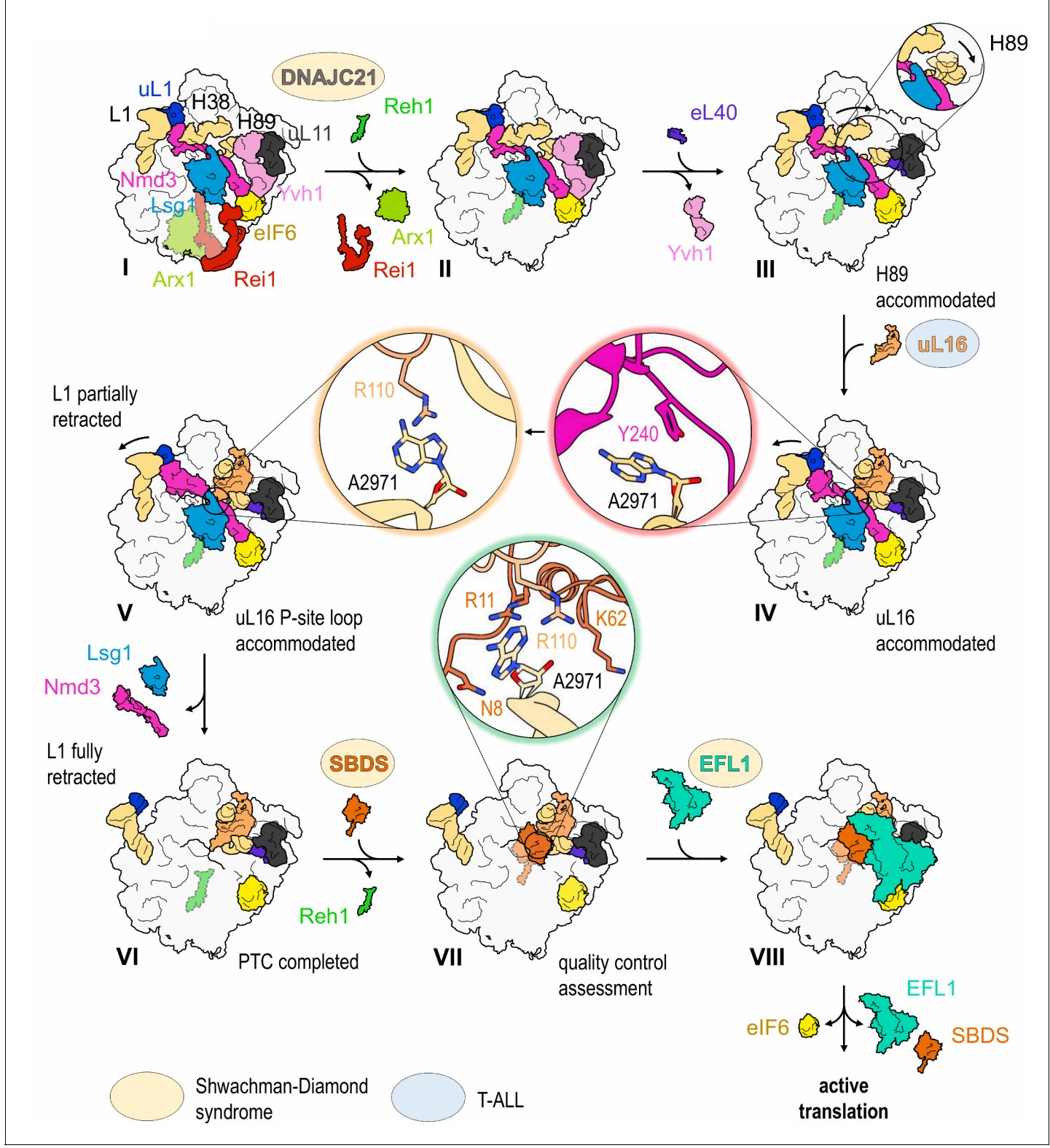

**Figure 7.** Mechanism of completion of PTC assembly and its quality control. Insets show the sequential interactions of base A2971 with Nmd3, uL16 and SBDS during PTC completion. Models for 60S-bound EFL1 and SBDS are based on EMD-3145, EMD-3146 and EMD-3147 (*Weis et al., 2015*). Proteins targeted by mutations in Shwachman-Diamond syndrome (yellow) and paediatric T-ALL (blue) are highlighted.

DOI: https://doi.org/10.7554/eLife.44904.019

The following figure supplement is available for figure 7:

*Figure 7 continued on next page*

*Figure 7 continued*

**Figure supplement 1.** The uL16 sequence and structure are highly conserved.
DOI: https://doi.org/10.7554/eLife.44904.020

events is consistent with the requirement for uL16 integration to allow SBDS binding both in vivo (*Weis et al., 2015*) and in vitro (*Sulima et al., 2014*) and with recent cryo-EM data (*Weis et al., 2015*; *Ma et al., 2017*; *Malyutin et al., 2017*). Importantly, we also note that the binding sites for SBDS, Nmd3 and Lsg1 are incompatible, precluding the removal of eIF6 before the departure of Lsg1 and Nmd3 (*Figure 6—figure supplement 2B*). We conclude that Nmd3 and Lsg1 are released prior to SBDS and EFL1 recruitment and that Lsg1 and Nmd3 function at least in part as placeholders for SBDS.

## Proofreading the newly assembled PTC

This study reinforces the concept that SBDS and EFL1 interrogate the structural and functional integrity of the completed PTC by licensing removal of the anti-association factor eIF6 (*Weis et al., 2015*) (*Figure 7*). The side chain of SBDS residue R11 and the backbone of K62 interact with the side chain of R110 within the uL16 P-site loop that stabilises the flipped-out conformation of base A2971 (*Figure 6—figure supplement 3A*). In addition, there are interesting parallels between the interactions of the uL16 P-site loop with the N-terminus of SBDS (pdb 6qkl) (*Weis et al., 2015*) and the contacts of uL16 with the P-site tRNA during decoding (pdb 5gak) (*Schmidt et al., 2016*) (*Figure 6—figure supplement 3B*).

The PET of the large ribosomal subunit appears to be directly monitored initially in the nucleus by Nog1 (*Wu et al., 2016*) and subsequently throughout the entire process of cytoplasmic assembly by Rei1 (*Greber et al., 2012*; *Greber et al., 2016*) and Reh1 (*Ma et al., 2017*) in turn. Indeed, the direct interaction between the α-helical domain of Lsg1 and the N-termini of both Rei1 and Reh1 (*Figure 1A*) suggests how events at the PET and PTC might be coupled during cytoplasmic maturation. Docking analysis raises the possibility that the C-terminus of Reh1 may interact directly with the N-terminus of SBDS during late 60S maturation (*Figure 6—figure supplement 3C*). However, the functional relevance of this observation remains to be tested. It is conceivable that Reh1 may not be removed from the exit tunnel until the pioneer round of translation, but the precise timing and mechanism of Reh1 release require elucidation.

## A coherent pathway targeted by mutations in leukaemia

Our data define a coherent pathway that is targeted by multiple mutations in sporadic and inherited forms of leukaemia (*Figure 7*). The association of the SDS clinical phenotype with mutations in several components of the 60S maturation pathway, including SBDS (*Boocock et al., 2003*), EFL1 (*Stepensky et al., 2017*) and DNAJC21 (*Tummala et al., 2016*; *Dhanraj et al., 2017*) provides compelling support for the hypothesis that SDS is a ribosomopathy (*Menne et al., 2007*; *Finch et al., 2011*). Importantly, given the high degree of conservation in the amino acid sequence and structure of the uL16 protein (*Figure 7—figure supplement 1A,B*), our cryo-EM data provide a mechanistic model that allows us to interpret the consequences of the recurrent uL16-R98S mutation found in paediatric T-ALL (*De Keersmaecker et al., 2013*). We propose that the uL16-R98S mutation may increase the flexibility of the P-site loop, reducing its ability to effectively compete with the Nmd3 eL22-like domain for the overlapping binding site at base A2971 in the PTC, thereby driving the equilibrium towards Nmd3 release.

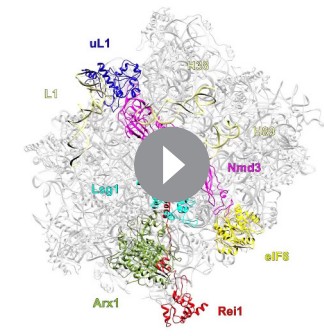

**Video 1.** Mechanism of completion of peptidyltransferase centre assembly.
DOI: https://doi.org/10.7554/eLife.44904.021

Our study suggests that Yvh1 dissociates from the pre-60S particle prior to the binding of eL40 and uL16 (*Figure 1A*). However, while this work was under review, Yvh1 was reported to bind cytoplasmic pre-60S particles carrying eL40 and uL16 (*Zhou et al., 2019*). Perhaps explaining this discrepancy, our study has exclusively analysed native pre-60S particles, while Zhou et al examined particles purified from Rlp24-mutant cells or following treatment with the inhibitor diazaborine (*Zhou et al., 2019*). Interpretation of the structural intermediates in the Zhou study should take also account of the heterogeneity in the deposited maps due to the classification strategy used.

In conclusion, we have used single-particle cryo-EM to demonstrate the conformational changes during cytoplasmic 60S subunit maturation that couple incorporation of the ribosomal proteins eL40 and uL16 to the release of Nmd3 in what is likely to be a universal mechanism in eukaryotes. Our data not only reveal how the central functional site of the ribosome is assembled, but provide a framework to interpret the consequences of mutations linked to leukaemia-associated ribosomopathies.

## Accession codes

The cryo-EM density maps have been deposited in the Electron Microscopy Data Bank with accession numbers EMD-10068, EMD-10071, EMD-4560, EMD-4636, EMD-4884 and EMD-4630. Atomic coordinates have been deposited in the Protein Data Bank, with entry codes 6RZZ, 6S05, 6QIK, 6QTZ, 6RI5 and 6QT0.

# Materials and methods

## SDS-PAGE and immunoblotting

Proteins present in crude extracts and purified pre-60S particles were separated on precast 4% to 12% NuPAGE gels (Novex life technologies) and blotted on a PVDF membrane (Carl Roth GmbH) using a tank-blot device (Hoefer). For immunoblotting, primary antibodies directed to pre-ribosome maturation factors or ribosomal proteins and a secondary goat anti-rabbit antibody (Sigma Aldrich) were used. All antibodies were described previously (*Zisser et al., 2018*; *Loibl et al., 2014*), except the eIF6 antibody that was purchased from GeneTex. Chemiluminescence signals were detected using the ChemiDoc Touch Imaging System (Bio-Rad) and the Clarity Western Blotting Detection Reagent as substrate.

## Purification of Lsg1-TAP pre-60S particles

For the isolation of late cytoplasmic pre-60S particles Lsg1-TAP (*Nissan et al., 2002*) was used as bait protein. Purification was performed in the absence of GTPase inhibitor. Affinity purification was performed using rabbit IgG covalently linked to magnetic beads as described (*Zisser et al., 2018*). Briefly, the Lsg1-TAP strain was grown in two litres of YPD complete medium (2% (w/v) peptone, 1% (w/v) yeast extract, 2% (w/v) glucose, 0.002% (w/v) adenine) to an $OD_{600}$ of 1.2 and cells were harvested by centrifugation for 2 min at 4000 x g. Cells were washed once in lysis buffer 1 (LB1; 20 mM HEPES, pH 7.5, 10 mM KCl, 2.5 mM $MgCl_2$, 1 mM EGTA, 0.5 mM PMSF, 1 mM DTT and FY-protease inhibitor (Serva). Cell lysis was performed in LB1 after addition of 1.5 volumes of glass beads by vigorous shaking for $4 \times 30$ s in a bead mill (Merkenschlager) with constant $CO_2$ cooling. After centrifugation for 30 min at 40,000 x g, the cleared lysates were loaded on 200 µl magnetic beads covalently coupled with rabbit IgG (*Oeffinger et al., 2007*) and incubated at 4°C for 90 min under constant mixing using an overhead rotator. After washing twice with 8 ml LB1 containing 1 mM DTT and once with 8 ml LB1 containing 100 mM NaCl and 1 mM DTT, beads were transferred to 0.5 ml reaction vials and pre-ribosomal particles eluted by overnight TEV cleavage in 150 µl LB1 containing 100 mM NaCl, 0.5 mM DTT and 2 µg of purified, RNAsin treated TEV protease. After removal of the resin by centrifugation for 5 min at 5000 x g, the pre-60S particles present in the supernatant were spotted onto grids and processed for cryo-EM freezing. SDS-PAGE and immunoblotting was used to monitor the quality of the isolated particles.

## Cryo-EM sample preparation

Pre-60 ribosomal TAP-tagged Lsg1 particles purified from *S. cerevisiae* were incubated 10 min at 4°C in the presence of 0.5% (v/v) glutaraldehyde (Sigma-Aldrich) to reduce preferential orientation and

DTT added to a final concentration of 6 mM. EM grids were prepared by adding 3 µL pre-60 ribosomal purified TAP-tagged Lsg1 particles (40 nM) to freshly glow-discharged Quantifoil R2/2 grids (PELCO easyGlow). Grids were blotted and flash frozen in liquid ethane at 100 K using a Vitrobot Mark IV (FEI Company).

## Cryo-EM data acquisition

Grids were screened on a Tecnai T12 microscope (FEI Company) and data acquisition performed under low-dose conditions on a Titan Krios microscope (FEI Company) operated at 300 kV over two sessions of ~70 hr each. The two datasets were recorded on a Falcon III detector (FEI Company) at a nominal magnification of 75,000x (effective pixel size of 1.065 Å on the object scale) with a defocus range of −0.8 to −3.2 µm and a dose of ~63 e−/Å$^2$. The acquisition of 7957 and 15,923 movies for each session was performed semi-automatically using EPU software (FEI Company).

## Cryo-EM image processing

Movies were corrected for the effects of beam-induced motion using MotionCor2 (*Zheng et al., 2017*). Contrast transfer function (CTF) parameters were estimated using GCTF (*Zhang, 2016*). All subsequent data processing was performed in RELION (*Scheres, 2012b*; *Scheres, 2012a*; *Kimanius et al., 2016*). Electron micrographs showing signs of drift or astigmatism were discarded, resulting in a dataset of 7279 and 12,261 images. A total of 696,991 (dataset 1) and 724,890 (dataset 2) particles were selected automatically in RELION. Extracted particles were subjected to two rounds of 2D and 3D classification to discard defective particles, resulting in 370,687 (dataset 1) and 512,903 particles (dataset 2). 3D auto-refinement resulted in an initial cryo-EM reconstruction with an overall resolution of 3.1 Å and 3.4 Å for the two datasets. After movie refinement and particle polishing the 'shiny' particles were subjected to further 3D auto-refinement and post-processing to yield maps with an overall resolution of 2.9 Å (dataset 1) and 3.2 Å (dataset 2) based on the gold-standard Fourier Shell Correlation (FSC) criterion calculated within RELION (*van Heel and Stöffler-Meilicke, 1985*; *Scheres and Chen, 2012*). However, the final maps were clearly heterogeneous in composition. We therefore sorted the images into subsets by a succession of 3D classifications using signal subtraction in RELION (*Penczek et al., 2006*) (*Figure 1—figure supplement 2*). A mask with a voxel value of one inside and zero outside extended by four pixels with a soft edge of ten pixels was applied to the intersubunit interface, containing Lsg1, Nmd3, eIF6, uL16, H38 and H89, was used for the first round of focused classification, providing 10 classes for each dataset (*Figure 1—figure supplement 2*, mask 1). In dataset 2, a mask was applied around the L1 stalk, uL1 and the OB, SH3 and eL31-like domains of Nmd3 to separate the 'open' and 'closed' L1 stalk conformations (*Figure 1—figure supplement 2*, mask 2). Finally, identical classes from both datasets were merged to improve the overall resolution. A mask was applied around the P stalk in states I and II to reveal different conformations of H89 and the presence of Yvh1 (*Figure 1—figure supplement 2*, mask 3). Similarly, further classification (using masks 2 and 3) was performed in states III and IV to reveal eL40 (State III, subclass 1) and the retracted L1-stalk (State V, subclass 1), respectively. Image processing converged to yield six distinct classes (states I-VI) with global resolutions ranging from 3.1 to 3.9 Å (*Figure 1—figure supplement 1E*). Local resolution was estimated to range from 2.3 to 6.3 Å using ResMap (*Kucukelbir et al., 2014*) (*Figure 1—figure supplement 1F*, *Figure 1—figure supplement 4*).

## Model building and refinement

As an initial starting model, the 3.0 Å crystal structure of the mature 60S subunit (*Ben-Shem et al., 2011*) (pdb 4v88) from *S. cerevisiae* was initially fitted as a rigid body into the cryo-EM map of state I using UCSF-Chimera (*Pettersen et al., 2004*). Atomic coordinates for Nmd3, Lsg1, eIF6 and uL1 (backbone atoms only) were taken from pdb code 5t62 (*Malyutin et al., 2017*); Rei1 and Arx1 from pdb code 5apn (*Greber et al., 2016*); Reh1 from pdb code 5h4p (*Ma et al., 2017*); eL40, uL16 and the mature conformation of H38 from pdb code 4v88 (*Ben-Shem et al., 2011*). Models were manually adjusted in Coot (*Emsley and Cowtan, 2004*) and further refined using Phenix (*Adams et al., 2010*) and REFMAC v5.8 adapted for EM-refinement (*Amunts et al., 2014*). Model evaluation was performed in MolProbity (*Chen et al., 2010*) (*Supplementary file 1A*). Cross-validation against over-fitting was performed as described (*Weis et al., 2015*) (*Figure 1—figure supplement 3*). Buried

surface areas were calculated using the *gmx sasa* routine in GROMACS (*Van Der Spoel et al., 2005*) using the double cubic lattice method (*Eisenhaber et al., 1995*) with a probe radius of 0.14 nm. Molecular visualization was performed in UCSF-Chimera (*Pettersen et al., 2004*), ChimeraX (*Goddard et al., 2018*), Pymol (The PyMOL Molecular Graphics System, Version 2.0.6 Schrödinger, LLC) and VMD (*Humphrey et al., 1996*).

## Molecular dynamics flexible fitting (MDFF)

Flexible fitting of atomic models was initially performed using MDFF (*Trabuco et al., 2008*). The system was set up *in vacuo* and subjected to energy minimization for 50,000 steps (50 ps) to relax any steric clashes. A production run of 1,000,000 steps (one ns) was followed to fit the atoms into the EM density. The magnitude of the forces applied to the atoms (scaling factor ξ) was adjusted to 0.3 kcal/mol. To prevent overfitting, harmonic restraints were applied to maintain the secondary structure with a force constant of 200 kcal mol$^{-1}$ rad$^{-2}$. Default values were used to restrain hydrogen bonds, *cis*-peptide bonds and chiral centres. All steps were performed using the VMD visualization tool (*Humphrey et al., 1996*). The model was optimized *in vacuo* using NAMD2 (*Phillips et al., 2005*) and the CHARMM36 force field (*Best et al., 2012*) for proteins and nucleic acids.

## Purification of Lsg1-TAP particles for XL-MS

Pre-ribosomal particles were purified from *S. cerevisiae* BY4741 cells by tandem affinity purification. Genomically expressed Lsg1-TAP was used as bait protein to isolate ribosome assembly intermediates. Therefore, 12 L YPD medium was inoculated from 300 ml overnight culture with an OD$_{600}$ of 0.1 and grown at 30°C to OD$_{600}$ = 0.8–1.0. Cells were harvested by centrifugation at 4300 x g and 4°C for 12 min. Cell pellets were resuspended in 80 ml cold lysis buffer (LB-P, 50 mM HEPES, pH 7.4, 100 mM KCl, 1.5 mM MgCl$_2$, 0.1% (v/v) NP-40, 5% (v/v) glycerol, pefabloc 1:100, aprotinin and leupeptin 1:1000) and centrifuged at 4000 x g and 4°C for 5 min. Washed cell pellets were resuspended in 20 ml LB-P and dripped into liquid nitrogen. Frozen droplets of cell suspension were stored at −80°C until milling in a pre-cooled Retsch ball mill MM400 at 30 Hz for 2 × 60 s. 150 ml ice cold LB-P was added to the frozen cell powder which was thawed on a rolling mixer at 4°C. Cell debris was separated from *S. cerevisiae* lysate by centrifugation at 30,000 x g and 4°C for 20 min. The lysate was incubated with 1.2 ml equilibrated IgG sepharose beads (GE Healthcare) at 4°C for 3 hr. IgG beads were washed 3x with LB-P and 1x with LB-DTT (50 mM HEPES, pH 7.4, 100 mM KCl, 1.5 mM MgCl$_2$, 0.1% (v/v) NP-40, 5% (v/v) glycerol, 1 mM DTT). IgG beads were loaded onto a 5 mL Polyprep column using 3 × 10 mL LB-DTT. The column was closed and IgG beads were incubated in 4.5 mL LB-DTT with 175 µl TEV protease (produced in-house, 1.5 µg/µL in 10% glycerol) at 4°C over night on a rolling incubator. IgG eluate was incubated with 1 mL equilibrated calmodulin affinity resin (Agilent) in 15 mL LB-CaCl$_2$ (50 mM HEPES, pH 7.4, 100 mM KCl, 1.5 mM MgCl$_2$, 0.02% (v/v) NP-40, 5% (v/v) glycerol, 2 mM CaCl$_2$) at a final CaCl$_2$ concentration of 2 mM on a rolling mixer at 4°C for 3 hr. Calmodulin beads were loaded onto a 5 mL Polyprep column using 2 × 20 mL LB-CaCl$_2$ and washed with 1 × 10 mL LB-CaCl$_2$. The column was closed and calmodulin beads were incubated with 550 µL LB-EGTA (50 mM HEPES, pH 7.4, 100 mM KCl, 1.5 mM MgCl2, 0.01% (v/v) NP-40, 5% (v/v) glycerol, 5 mM EGTA) for 20 min at 4°C on a rolling incubator. The eluate was collected and the elution was repeated 3x with 450 µL LB-EGTA. Eluates 1–4 were concentrated using an Amicon Ultra 10 K 0.5 mL filter (Merck Millipore) to a final volume of ca. 100 µl in crosslinking buffer (20 mM HEPES, pH 8.3, 5 mM MgCl$_2$).

## Chemical crosslinking coupled to mass spectrometry (XL-MS)

XL-MS was carried out essentially as described (*Leitner et al., 2014*). In short, roughly 100 µg of eluate was directly cross-linked with 1.5 mM disuccinimidyl suberate d0/d12 (DSS, Creativemolecules Inc), digested with trypsin and subsequently enriched for cross-linked peptides. LC-MS/MS analysis was carried out on an Orbitrap Fusion Tribrid mass spectrometer (Thermo Electron, San Jose, CA). Data were searched using *xQuest* in iontag mode against a database containing ribosomal proteins and known assembly factors (total of 380 proteins) of *S. cerevisiae* with a precursor mass tolerance of 10 ppm. For each experiment, only unique cross-links were considered and only high-confidence cross-linked peptides that were identified with a delta score (deltaS) below 0.95 and an Id-Score

above 32, translating to an FDR of 0.2 (*Erzberger et al., 2014*), were selected for this study. Cross-links were visualised by xiNET software (*Combe et al., 2015*).

## NMR spectroscopy

All NMR data was collected at 298 K on an Avance II + 700 MHz spectrometer, equipped with a cryogenic triple-resonance TCI probe. 2D BEST-Trosy and standard 3D triple-resonance experiments were acquired with a sample of 100 µM $^{15}$N,$^{13}$C labelled *A. fulgidus* Nmd3 (residues 22–150) in PBS buffer with 1.5 mM DTT at pH 7.2. Data were processed using Topspin 3.0 (Bruker) and analysed using SPARKY (T. D. Goddard and D. G. Kneller - University of California, San Francisco).

## Plasmids

PCR was used to amplify the coding sequence for wild-type uL16 and Nmd3 plus 500 bp of upstream and downstream of the coding sequence using yeast genomic DNA as template. PCR products were cloned into vectors pRS316 (*CEN, URA*) and pRS313 (*CEN, HIS*) using NEBuilder HiFi DNA Assembly Master Mix (New England Biolabs). Partially overlapping primers containing the mutation were used to perform site-directed mutagenesis. For plasmids and primers, see *Supplementary files 3A, B*.

## Genetic complementation assays

Haploid yeast cells (strain NE0206, see *Supplementary file 3C*) transformed with plasmids expressing wild type or mutant Nmd3 or vector alone were spotted in ten-fold serial dilutions onto solid synthetic defined -Ura -His medium containing glucose as carbon source for 2 days at 37 ˚C.

## Acknowledgements

We thank Micheline Fromont-Racine, Ed Hurt, Arlen W Johnson, Sabine Rospert, John Woolford, Bernard L Trumpower and Juan P Ballesta for generous gifts of antibodies and strains. We acknowledge Diamond for access and support of the Cryo-EM facilities at the UK national electron bio-imaging centre (eBIC), (proposal EM17057), funded by the Wellcome Trust, MRC and BBSRC; the Astbury Biostructure Laboratory, in particular Dr. Rebecca Thompson, for cryo-EM data collection funded by the University of Leeds and the Wellcome Trust (108466/Z/15/Z); Dr. Dimitri Y Chirgadze for data collection at the Cryo-EM Facility, Department of Biochemistry, University of Cambridge, funded by the Wellcome Trust (206171/Z/17/Z; 202905/Z/16/Z), the Departments of Biochemistry and Chemistry, the Schools of Biological Sciences and Clinical Medicine and the University of Cambridge; the European NMR Large-Scale Facility (Utrecht, Holland); J Wilson for help with computing and D Merla for technical support. The work was supported by a Specialist Programme from Bloodwise (12048), the UK Medical Research Council (MC_U105161083, to AJW), the Austrian Science foundation FWF Grants (P26136 and P29451, to HB), a Wellcome Trust strategic award to the Cambridge Institute for Medical Research (100140), a core support grant from the Wellcome Trust and MRC to the Wellcome Trust-Medical Research Council Cambridge Stem Cell Institute, the Connor Wright Project and the Cambridge National Institute for Health Research Biomedical Research Centre. FS acknowledges funding from the German Science Foundation Emmy Noether Programme (STE 2517/1–1) and Collaborative Research Centre (SFB) 969, Project A06.

## Additional information

### Funding

| Funder | Grant reference number | Author |
| --- | --- | --- |
| Medical Research Council | MC_U105161083 | Alan John Warren |
| Bloodwise | 12048 | Alan John Warren |
| Wellcome | 108466/Z/15/Z | Edwin Chen |
| Deutsche Forschungsgemeinschaft | Emmy Noether Programme STE 2517/1-1 | Florian Stengel |

| Deutsche Forschungsge-meinschaft | Collaborative Research Center 969 Project A06 | Florian Stengel |
|---|---|---|
| Fonds zur Förderung der Wis-senschaftlichen Forschung | FWF Grants P26136 | Helmut Bergler |
| Fonds zur Förderung der Wis-senschaftlichen Forschung | FWF Grants P29451 | Helmut Bergler |

The funders had no role in study design, data collection and interpretation, or the decision to submit the work for publication.

## Author contributions

Vasileios Kargas, Formal analysis, Investigation, Methodology, Writing—original draft, Writing—review and editing, Cryo-EM data acquisition, Data processing, Model building, Structure validation; Pablo Castro-Hartmann, Formal analysis, Investigation, Methodology, Writing—original draft, Cryo-EM data acquisition, Data processing; Norberto Escudero-Urquijo, Formal analysis, Investigation, Writing—original draft, Writing—review and editing, Cryo-EM grid preparation, Data collection, Yeast genetics; Kyle Dent, Formal analysis, Methodology, Cryo-EM data processing and structure determination; Christine Hilcenko, Writing—original draft, Writing—review and editing, Protein expression and NMR spectroscopy; Carolin Sailer, Formal analysis, Investigation, Methodology, Writing—original draft, Writing—review and editing, Crosslinking mass spectrometry data acquisition and analysis; Gertrude Zisser, Methodology, Writing—original draft, Writing—review and editing, Sample preparation; Maria J Marques-Carvalho, Investigation, Methodology, Writing—review and editing, Cryo-EM sample preparation, Data processing, Model building; Simone Pellegrino, Investigation, Writing—original draft, Writing—review and editing, Model building; Leszek Wawiórka, Investigation, Writing—original draft, Writing—review and editing, Yeast genetics; Stefan MV Freund, Conceptualization, Supervision, Investigation, Methodology, Writing—original draft, Writing—review and editing, NMR spectroscopy; Jane L Wagstaff, Conceptualization, Validation, Investigation, Methodology, Writing—original draft, Writing—review and editing, NMR sprectroscopy; Antonina Andreeva, Formal analysis, Investigation, Writing—original draft, Writing—review and editing; Alexandre Faille, Data curation, Formal analysis, Validation, Methodology, Writing—original draft, Writing—review and editing, Data acquisition; Edwin Chen, Resources, Data curation, Investigation; Florian Stengel, Conceptualization, Data curation, Formal analysis, Supervision, Validation, Methodology, Writing—original draft, Writing—review and editing; Helmut Bergler, Formal analysis, Supervision, Investigation, Methodology, Writing—original draft, Writing—review and editing; Alan John Warren, Conceptualization, Data curation, Formal analysis, Supervision, Funding acquisition, Investigation, Methodology, Writing—original draft, Project administration, Writing—review and editing, model building

## Author ORCIDs

Vasileios Kargas https://orcid.org/0000-0001-8588-7285
Norberto Escudero-Urquijo https://orcid.org/0000-0002-8201-5884
Carolin Sailer https://orcid.org/0000-0001-7735-6070
Edwin Chen http://orcid.org/0000-0003-0742-9734
Florian Stengel https://orcid.org/0000-0003-1447-4509
Helmut Bergler http://orcid.org/0000-0002-7724-309X
Alan John Warren https://orcid.org/0000-0001-9277-4553

## Decision letter and Author response

Decision letter https://doi.org/10.7554/eLife.44904.061
Author response https://doi.org/10.7554/eLife.44904.062

# Additional files

## Supplementary files

• Supplementary file 1. Data collection, model refinement and validation (**A**), and summary of modelled ribosomal proteins, assembly factors and rRNA (**B**).
DOI: https://doi.org/10.7554/eLife.44904.022

• Supplementary file 2. Crosslinks identified by XL-MS. Data for two independent immunoprecipitation experiments (**A**, **B**), including the exact amino acid sequence of the cross-linked peptides and the position of the cross-linked lysine residue 'crosslinked peptide', the name of the respective protein 'protein1' and 'protein2', nature of the cross-link 'type', the absolute position of the cross-linked lysine residues within the UniProt or construct sequence 'Absolute Position 1' and 'Absolute Position 2'. 'deltaS' gives the delta score of the respective crosslink and is a measure of how close the best assigned hit was scored in regard to the second best, 'Id-Score', which is a weighted sum of different scores used to assess the quality of the composite MS2 spectrum as calculated by xQuest.
DOI: https://doi.org/10.7554/eLife.44904.023

• Supplementary file 3. Plasmids (**A**), primers (**B**), and yeast strains (**C**).
DOI: https://doi.org/10.7554/eLife.44904.024

• Transparent reporting form
DOI: https://doi.org/10.7554/eLife.44904.025

## Data availability

The cryo-EM density maps have been deposited in the Electron Microscopy Data Bank with accession numbers EMD-10068, EMD-10071, EMD-4560, EMD-4636, EMD-4884 and EMD-4630. Atomic coordinates have been deposited in the Protein Data Bank, with entry codes 6RZZ, 6S05, 6QIK, 6QTZ, 6RI5 and 6QT0.

The following datasets were generated:

| Author(s) | Year | Dataset title | Dataset URL | Database and Identifier |
|---|---|---|---|---|
| Vasileios Kargas, Pablo Castro-Hartmann, Norberto Escudero-Urquijo, Kyle Dent, Alan John Warren | 2019 | Cytoplasmic 60S ribosomal subunit (state I - subclass 1) | http://www.ebi.ac.uk/pdbe/entry/emdb/EMD-10066 | Electron Microscopy Data Bank, EMD-10066 |
| Vasileios Kargas, Pablo Castro-Hartmann, Norberto Escudero-Urquijo, Kyle Dent, Alan John Warren | 2019 | Cytoplasmic 60S ribosomal subunit (state I - subclass 2) | http://www.ebi.ac.uk/pdbe/entry/emdb/EMD-10067 | Electron Microscopy Data Bank, EMD-10067 |
| Vasileios Kargas, Pablo Castro-Hartmann, Norberto Escudero-Urquijo, Kyle Dent, Alan John Warren | 2019 | Cytoplasmic 60S ribosomal subunit (state II - subclass 1) | http://www.ebi.ac.uk/pdbe/entry/emdb/EMD-10070 | Electron Microscopy Data Bank, EMD-10070 |
| Vasileios Kargas, Pablo Castro-Hartmann, Norberto Escudero-Urquijo, Kyle Dent, Alan John Warren | 2019 | Cytoplasmic 60S ribosomal subunit (state III - subclass 1) | http://www.ebi.ac.uk/pdbe/entry/emdb/EMD-10009 | Electron Microscopy Data Bank, EMD-10009 |
| Vasileios Kargas, Pablo Castro-Hartmann, Norberto Escudero-Urquijo, Kyle Dent, Alan John Warren | 2019 | Cytoplasmic 60S ribosomal subunit (state V - subclass 1) | http://www.ebi.ac.uk/pdbe/entry/emdb/EMD-10039 | Electron Microscopy Data Bank, EMD-10039 |
| Kargas V, Castro PH, Escudero NU, | 2019 | Cytoplasmic 60S ribosomal subunit (state I) | http://www.ebi.ac.uk/pdbe/entry/emdb/EMD- | Electron Microscopy Data Bank, EMD-100 |

| | | | | |
|---|---|---|---|---|
| Dent K, Warren AJ | | | 10068 | 68 |
| Kargas V, Castro PH, Escudero NU, Dent K, Warren AJ | 2019 | Cytoplasmic 60S ribosomal subunit (state II) | http://www.ebi.ac.uk/pdbe/entry/emdb/EMD-10071 | Electron Microscopy Data Bank, EMD-10071 |
| Kargas V, Castro PH, Escudero NU, Dent K, Warren AJ | 2019 | Cytoplasmic 60S ribosomal subunit (state III) | http://www.ebi.ac.uk/pdbe/entry/emdb/EMD-4560 | Electron Microscopy Data Bank, EMD-4560 |
| Kargas V, Castro PH, Escudero NU, Dent K, Warren AJ | 2019 | Cytoplasmic 60S ribosomal subunit (state IV) | http://www.ebi.ac.uk/pdbe/entry/emdb/EMD-4636 | Electron Microscopy Data Bank, EMD-4636 |
| Kargas V, Castro PH, Escudero NU, Dent K, Warren AJ | 2019 | Cytoplasmic 60S ribosomal subunit (state V) | http://www.ebi.ac.uk/pdbe/entry/emdb/EMD-4884 | Electron Microscopy Data Bank, EMD-4884 |
| Kargas V, Castro PH, Escudero NU, Dent K, Warren AJ | 2019 | Cytoplasmic 60S ribosomal subunit (state VI) | http://www.ebi.ac.uk/pdbe/entry/emdb/EMD-4630 | Electron Microscopy Data Bank, EMD-4630 |
| Kargas V, Castro PH, Escudero NU, Dent K, Warren AJ | 2019 | Atomic model of cytoplasmic 60S ribosomal subunit (state I) | http://www.rcsb.org/structure/6RZZ | Protein Data Bank, 6RZZ |
| Kargas V, Castro PH, Escudero NU, Dent K, Warren AJ | 2019 | Atomic model of cytoplasmic 60S ribosomal subunit (state II) | http://www.rcsb.org/structure/6S05 | Protein Data Bank, 6S05 |
| Kargas V, Castro PH, Escudero NU, Dent K, Warren AJ | 2019 | Atomic model of cytoplasmic 60S ribosomal subunit (state III) | http://www.rcsb.org/structure/6QIK | Protein Data Bank, 6QIK |
| Kargas V, Castro PH, Escudero NU, Dent K, Warren AJ | 2019 | Atomic model of cytoplasmic 60S ribosomal subunit (state IV) | http://www.rcsb.org/structure/6QTZ | Protein Data Bank, 6QTZ |
| Kargas V, Castro PH, Escudero NU, Dent K, Warren AJ | 2019 | Atomic model of cytoplasmic 60S ribosomal subunit (state V) | http://www.rcsb.org/structure/6RI5 | Protein Data Bank, 6RI5 |
| Kargas V, Castro PH, Escudero NU, Dent K, Warren AJ | 2019 | Atomic model of cytoplasmic 60S ribosomal subunit (state VI) | http://www.rcsb.org/structure/6QT0 | Protein Data Bank, 6QT0 |

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
