## [Decision Letter]

Thank you for submitting your article "Cytoplasmic 60S maturation landscape corrupted in leukemia." for consideration by *eLife*. Your article has been reviewed by two peer reviewers, and the evaluation has been overseen by a Reviewing Editor and John Kuriyan as the Senior Editor. The following individuals involved in review of your submission have agreed to reveal their identity: Sebastian Klinge (Reviewer #1); Alexey Amunts (Reviewer #2).

The reviewers have discussed the reviews with one another and the Reviewing Editor has drafted this decision to help you prepare a revised submission.

Summary:

The authors used cryo-electron microscopy successfully to demonstrate the conformational changes during cytoplasmic 60S maturation that engender the release of Nmd3 and the incorporation of uL16.

Essential revisions:

The two reviewers are in agreement that the paper presents valuable data that advance the field of 60S ribosome assembly, and that the paper merits publication in *eLife* if the problems of presentations are rectified. Because the reviewers pointed out many shortcomings, the Reviewing Editor left the individual reviews intact below.

Most importantly, refined coordinate files of all states should be made available for the scientific community. Also, one of the reviewers after writing his/her review commented that a similar cryo-EM study just appeared in press (Zhou et al., in Nature Comm. Nature 10, Article number: 958 (https://www.nature.com/articles/s41467-019-08880-0#data-availability)), where Nmd3-TAP and uL16∆C-TAP were used for pull down. In general, the latter study is consistent with the current one. However, there is a discrepancy worth considering addressing: Zhou et al. particularly focus on the mechanism of how Yvh1 releases Mrt4 that is required for P stalk assembly. From the text, it appears that Yvh1 is bound to pre-60S containing eL40. While the coordinates have not been released, it appears to contradict the data in this paper that suggest that Yvh1 and eL40 are mutually exclusive, particularly in state III. Thus, the authors should clarify this point and perhaps other discrepancies.

Reviewer #1:

Here Kargas et al. have elucidated the mechanistic steps involved in cytoplasmic 60S maturation steps. The authors' elegant use of cryo-electron microscopy and their scholarly interpretation have shown that a series of conformational changes are fine-tuned to allow for the release of Nmd3 and the incorporation of uL16. The reported results are of high quality and should be published in *eLife* provided that the following points are addressed.

1) The cross-linking mass spectrometry data is quoted as a means of delineating a time course of protein-protein interaction. While the type of interactions seen for Rei1 and Reh1 with other factors is clearly indicative for mutually exclusive interactions, there are also numerous crosslinks that do not make immediate sense. Since only a single protein bait was used for the analyzed samples (Lsg1), numerous small subunit assembly factors are present in this sample, which have been cross-linked to large subunit assembly factors (Figure 1—figure supplement 5). These include Esf1, Sas10, Nan1, Enp2, Enp1, Utp30, Rrp7, Rrp9, Bud21, Utp6, Rrp5, Esf1, Puf6, Utp15, Utp4, Tsr3, and Enp2. Since the crosslinking and mass spectrometry data has had a supportive role for the interpretation of the data, it would therefore be very helpful for the authors to comment on the validity of all observed crosslinks and/or their limitations.

2) State VI lacks Nmd3 and Lsg1, the bait protein, and the authors argue (Results paragraph one) that these particles have continued to mature following the purification. An alternative interpretation would be that Lsg1 and Nmd3 have dissociated from a subpopulation of particles in which their affinity for the pre-60S particles is relatively low already. While this would not majorly change the authors' model, it may highlight a technical limitation of sample preparation. The authors' comments about this possibility would be very helpful.

3) In Figure 1A, low-pass filtered density for eL40 is shown in the composite image of State III. Since I was unable to see eL40 in the map of State III, did the authors mask out this protein for this reconstruction or was it inadvertently incorporated in State III?

4) Within the pdbs of states I-VI zinc ions are missing in several ribosomal proteins (including eL40) as well as the two zinc ions shown in Figure 2A, which belong to Nmd3. Given the high quality of these reconstructions and their ability to serve as future structural references, adding zinc ions would be very helpful for the scientific community.

5) Since there are six different states, it would be helpful to indicate in each figure, from which state a particular view originates. With this information the reader will be able to look at the correct pdb and EM map at the same time.

6) Local resolution is an important factor to consider for the interpretation of all assembly factors involved in this study. While Figure 1—figure supplement 1F shows a distribution of resolutions, it would be helpful to have a supplementary figure to Figure 1A to show local resolutions for all discussed ribosome assembly factors. These resolution estimations could be visualized using the latest features of Relion3 where low-pass filtering is also applied to these regions at their respective local resolution. Along similar lines, figures that show side chain and bases should be backed up with corresponding supplementary figures showing the corresponding density – ideally not low-pass filtered.

Reviewer #2:

Several models of pre-60S ribosome assemblies have been proposed recently in the literature. The most comprehensive studies focused on the early nuclear pre-60S maturation step by cryo-EM (Kater et al., 2017; Sanghai et al., 2018). The authors of the submitted manuscript have a long-standing interest in how the maturation process is affected in Shwachman-Diamond syndrome, and they published a series of related studies, including solid structural work explaining the mechanism of late eIF6 release from the nascent 60S ribosomal subunit (Weis et al., 2015). In the current work they further expand on that direction and characterize the 60S ribosomal subunit late maturation intermidiate using cryo-EM and cross-linking mass spectrometry. Particularly, the study describes characterization of *S. cerevisiae* pre-60S isolated using Lsg1-TAP as a bait. Intense computational analysis then resulted in six reconstructions at 3.1-3.9 Å resolution. For two of the reconstructions, namely states I and III, the models have been built and refined. The authors then analyzed the structural information and identified several putative consequent key steps in the assembly of the yeast 60S, leading to the conclusion that the incorporation of uL16 causes the release of the factor Nmd3, which is an important step in the maturation of the peptidyltransferase center. Since uL16 of human ribosomes is mutated in T-cell acute lymphoblastic leukemia, the authors propose a mechanism of an increased flexibility of the P-site loop that reduces its ability to effectively compete with the Nmd3.

While the study provides a comprehensive analysis of the late assembly of the yeast 60S and offers important fundamental insight for the ribosomal community, it is equally important that the quality of the presentation will match. That is not the case at the moment, which makes it difficult to follow some of the fine structural details. I would therefore suggest revising the presentation of the data so that the paper can be considered for a publication. For the revised version, I would also recommend adding line numbers so that it is easier for the authors to follow reviewers' comments.

Overall, not being an expert in the field of the 60S assembly, the description of the structural data sounds convincing, sensible and most detailed. It is also consistent with the previously published literature, and the conclusions regarding the PTC assembly are likely to be universal, and very interesting.

Specific comments regarding the presentation of the data:

– Abstract: The Abstract should be designed in the following way: introduction, main results, conclusion, importance. The first sentence is related to the importance and should be placed accordingly. In the second sentence, 'entire dynamic landscape' should be removed, because authors don't really know the entire landscape. It is also certainly not 'dynamic' because only individual snapshots are presented by cry-EM. Last sentence, the data concerns yeast ribosome, therefore leukemia related effects can only be suggested, whereas if authors are interested in 'illuminating', they should have worked with the relevant mutants of human ribosomes.

– Results and Figures:

While six high resolution maps were obtained in this study (states I-VI), only two models were built (states I and III), and some of the comparison is done on the level of the density maps. I consider it as a problem because it limits readers' accessibility to the reported information. There are certain basic standards in structural biology implying that the experimental data (density maps) are interpreted into models (atomic coordinates), which can be validated according to the tools originally developed for X-ray crystallography, the validated models are then used to discuss the molecular mechanisms. Here authors compromise those standards by skipping validation of the model building for states II, IV, V, VI or at least it is not reported in the current version of the manuscript. The authors should build the models, validate all of them, upload all the structural information to the public domain and present the validation details in the supplementary materials. Currently, Supplementary file 1A shows data from two models, whereas Supplementary file 1B shows data for a single model, the rest is missing.

All the figures should have consistent representation of the densities and the models. For example in Figure 2C and 2D the density is shown in two different modes and not in the same colors.

Figure 4A and 4D describes flexibility if H89. How can an atomic model from one study be compared with experimental density from another study? Please compare density with density or model with model.

Also for Figure 5; in 5A-C, it is the overall models that should be compared, especially given that the authors claim near atomic structures; in the bottom insets, the density should be shown in blue mesh, which is the standard for high resolution work and provides evidence of experimental data. The same goes for 5E, the current representation is not possible to track.

– Discussion:

There is quite a gap between reporting a structure of yeast ribosome to discussing a mechanism related to inherited forms of leukemia. The association of the clinical phenotype is certainly important, but such an implication should be justified and conservation between yeast and human ribosomes should be illustrated in the supplementary material.

---

## [Author Response]

Essential revisions:The two reviewers are in agreement that the paper presents valuable data that advance the field of 60S ribosome assembly, and that the paper merits publication in eLife if the problems of presentations are rectified. Because the reviewers pointed out many shortcomings, the Reviewing Editor left the individual reviews intact below.Most importantly, refined coordinate files of all states should be made available for the scientific community. Also, one of the reviewers after writing his/her review commented that a similar cryo-EM study just appeared in press (Zhou et al., in Nature Comm. Nature 10, Article number: 958 (https://www.nature.com/articles/s41467-019-08880-0#data-availability)), where Nmd3-TAP and uL16∆C-TAP were used for pull down. In general, the latter study is consistent with the current one. However, there is a discrepancy worth considering addressing: Zhou et al. particularly focus on the mechanism of how Yvh1 releases Mrt4 that is required for P stalk assembly. From the text, it appears that Yvh1 is bound to pre-60S containing eL40. While the coordinates have not been released, it appears to contradict the data in this paper that suggest that Yvh1 and eL40 are mutually exclusive, particularly in state III. Thus, the authors should clarify this point and perhaps other discrepancies.

All the refined coordinate files have been deposited in EMDB and in the PDB for the scientific community.

With respect to the recently published study by Zhou et al. (Zhou et al., 2019), we agree that there are indeed some discrepancies with our study. There are a number of potential explanations for this. Importantly, we would like to emphasise that our study has focused entirely on the characterisation of purified native pre-60S particles. By contrast, the study by Zhou et al. examined particles purified from Rlp24 mutant cells or following the partial arrest of ribosome biogenesis with the chemical inhibitor diazaborine. The possibility that ribosome assembly may have been perturbed in the Zhou et al. study must therefore be considered when interpreting their structural intermediates. In addition, examination of the maps deposited by Zhou et al. shows apparent particle orientation bias and heterogeneity in the map densities. Indeed, in the online peer review document for their manuscript, Zhou et al. state that they “did not attempt to reconstruct factors at the exit tunnel, as densities at the exit tunnel were heterogeneous because of our classification approach”. In our study we felt that it was critical to rigorously classify the entire ribosomal subunit to allow us to draw what we believe to be robust conclusions about the structural intermediates we have identified.

Specific points with respect to the study by Zhou et al:

1) The map corresponding to state ‘RI’ in Figure 1 has density for uL16, but also appears to have clear density for Arx1 which should be absent when uL16 is bound. There is also heterogeneity in the conformation of H38, which is present in a mixture of ‘closed’ and ‘open’ conformations.

2) The density assigned to Lsg1 that lies over the surface of Tif6 in the particles represented in Figures 1E, F also appears to be present in the map corresponding to Figure 1D upon low pass filtering, but Lsg1 is clearly absent. We suggest that this density has been misassigned as an extension of Lsg1 (see also Ma et al., 2017). Instead, our data indicate that this density corresponds to the N-terminus of Rei1 (see Figure 1, state 1).

3) Zhou et al., Figure 1D is said to represent a “Pre-Lsg1” particle. However, it seems more likely to us that this represents a class from which Lsg1 has dissociated. Examination of the corresponding deposited map reveals that the density occupying the exit tunnel is likely Reh1, which we can clearly distinguish on the basis of specific side-chains such as Reh1 Y394 (G355 in Rei1), M401 (V362 in Rei1) and the presence of a distinct helical domain extending from residues R376-E391. The presence of Reh1 in the map corresponding to Figure 1D would be inconsistent with its assignment as an early “Pre-Lsg1” state. In fact, we also observed a similar intermediate in our own dataset (see subsection “Loading of eL40 stabilizes H89 to facilitate uL16 recruitment”) upon sub-classifying state III in which Nmd3 is rotated towards H89. However, Lsg1 is absent, Reh1 is in the PET and the particle lacks Arx1, Yvh1 and uL16. We conclude that this intermediate represents a non-physiological state that is more likely to be a consequence of Lsg1 dissociation and not a “Pre-Lsg1” particle.

4) The large size of our primary dataset allowed us to perform extensive focused classification that we believe has enabled us to resolve the marked compositional heterogeneity of the late cytoplasmic pre-60S particles. In addition, our data are also consistent with recent quantitative mass spectrometry and genetic results deposited in the bioRxiv (Nerurkar et al. 2019, doi: https://doi.org/10.1101/462333).

5) To address the editor’s comments, we have added the following text to our

manuscript:

“Our study suggests that Yvh1 dissociates from the pre-60S particle prior to the binding of eL40 and uL16 (Figure 1A). However, while this work was under review, Yvh1 was reported to bind cytoplasmic pre-60S particles carrying eL40 and uL16 (Zhou et al., 2019). Perhaps explaining this discrepancy, our study has exclusively analysed native pre-60S particles, while Zhou et al. examined particles purified from Rlp24-mutant cells or following treatment with the inhibitor diazaborine (Zhou et al., 2019). Interpretation of the structural intermediates in the Zhou study should also take account of the heterogeneity in the deposited maps due to the classification strategy used.”

Reviewer #1:Here Kargas et al. have elucidated the mechanistic steps involved in cytoplasmic 60S maturation steps. The authors' elegant use of cryo-electron microscopy and their scholarly interpretation have shown that a series of conformational changes are fine-tuned to allow for the release of Nmd3 and the incorporation of uL16. The reported results are of high quality and should be published in eLife provided that the following points are addressed.1) The cross-linking mass spectrometry data is quoted as a means of delineating a time course of protein-protein interaction. While the type of interactions seen for Rei1 and Reh1 with other factors is clearly indicative for mutually exclusive interactions, there are also numerous crosslinks that do not make immediate sense. Since only a single protein bait was used for the analyzed samples (Lsg1), numerous small subunit assembly factors are present in this sample, which have been cross-linked to large subunit assembly factors (Figure 1—figure supplement 5). These include Esf1, Sas10, Nan1, Enp2, Enp1, Utp30, Rrp7, Rrp9, Bud21, Utp6, Rrp5, Esf1, Puf6, Utp15, Utp4, Tsr3, and Enp2. Since the crosslinking and mass spectrometry data has had a supportive role for the interpretation of the data, it would therefore be very helpful for the authors to comment on the validity of all observed crosslinks and/or their limitations.

While we have demonstrated repeatedly that we are able to detect cross-links and protein-protein interactions with high confidence using the approach we have applied in this study (see for example Patel, Nature Commun, 2017; Erzberger et al., 2014; Leitner, Trends Biochem Sci. 2016), an extensive previous study from us also showed that useful information can be lost if too high cut-offs are used (Erzberger et al., 2014). We have therefore used settings and MS cut-offs for this study that translate to an expected FDR of roughly 20 percent for the identified cross-links. Thus, the overall list of detected cross-links will also contain false-positives, being a likely explanation for some of the crosslinks that were detected to small subunit assembly factors within this sample. Nonetheless, the links that are discussed in detail in the manuscript (Rei1: Arx1; Rei1 to eL24 and Reh1 to eL24), we do see in both independent samples and with even higher cut-offs (Id >36) or with multiple independent peptides, and are thus particularly trustworthy.

Furthermore, the crosslinking mass spectrometry (XL-MS) data for this study was obtained from two independent epitope-tagged Lsg1 sample preparations that were used as bait. As such, any delineation of a time-course or 60S subunit maturation has to be indirect, as the reviewer correctly points out. However, as XLMS is an ensemble technology, this means that in principle crosslinks from all distinct populations that were present in the sample will be detected. Thus, the list of detected cross-links will likely contain cross-links from the various states obtained by cryo-EM and in combination these techniques are then indeed able to generate valid information on the subunit maturation process of the 60S particle, as stated in the manuscript.

As written in the Materials and methods section, we have deliberately decided against a narrow search using only known 60S assembly factors, but have conducted a search against all ribosomal proteins and known assembly factors in *S. cerevisiae* (total of 380 proteins) in order to obtain a more general view on all populations that were present in the sample. The obtained crosslinks on numerous small subunit assembly factors will therefore likely stem from particles that were present in the sample due to the biochemical enrichment process and that were at the same time not numerous enough to constitute one of the distinct states detected by cryo-EM. In order to clarify above points we have added the following clarification and reference to the manuscript:

“For each experiment, only unique crosslinks were considered and only cross-linked peptides that were identified with a delta score (deltaS) below 0.95 and an Id-Score above 32, translating to an FDR of 0.2 (Erzberger et al., 2014), were selected for this study.”

2) State VI lacks Nmd3 and Lsg1, the bait protein, and the authors argue (Results paragraph one) that these particles have continued to mature following the purification. An alternative interpretation would be that Lsg1 and Nmd3 have dissociated from a subpopulation of particles in which their affinity for the pre-60S particles is relatively low already. While this would not majorly change the authors' model, it may highlight a technical limitation of sample preparation. The authors' comments about this possibility would be very helpful.

As the reviewer correctly points out, we cannot exclude the possibility that Lsg1 and Nmd3 may simply have dissociated from the state V particles due to low affinity, reflecting a potential technical limitation in the sample preparation. On the other hand, if this were the case, we might have expected to also identify an additional intermediate downstream of state V lacking Lsg1 but retaining Nmd3. Indeed, as we state in subsection “Loading of eL40 stabilizes H89 to facilitate uL16 recruitment”, focused classification of state III identified a population of particles in which Lsg1 is absent, but Nmd3 is retained but with its eL31-like domain rotated towards H89. We suggest that this represents a nonphysiological state rather than a “Pre-Lsg1” particle given that Reh1 is present in the PET and Arx1 is absent from these particles. This state may correspond to the class depicted by Zhou et al. in Figure 1D of their manuscript as a ‘Pre-Lsg1’ particle. Importantly, however, we agree with the reviewer that the possible dissociation of Nmd3/Lsg1 after state V does not significantly alter the conclusions that we wish to draw.

We have modified the text to clarify these points:

Results section:

“The ability to capture state VI (lacking both Nmd3 and Lsg1) likely reflects ongoing maturation of the particles or possibly the dissociation of Lsg1 and Nmd3 during immunopurification.”

“Although focused classification of state III identified a subset of particles in which the Nmd3 eL31-like domain is rotated towards H89, this class lacks Lsg1 and Arx1, with Reh1 present in the PET. We suggest that the rotated conformation of the Nmd3 eL31-like domain in this subset of state III particles is a consequence of Lsg1 dissociation during sample preparation rather than a physiologically relevant “pre-Lsg1” state.”

3) In Figure 1A, low-pass filtered density for eL40 is shown in the composite image of State III. Since I was unable to see eL40 in the map of State III, did the authors mask out this protein for this reconstruction or was it inadvertently incorporated in State III?

We apologise for the confusion. We have deposited an additional map “State III, subclass 1” which clearly shows the density for eL40. We have added the following sentence in subsection “Cryo-EM image processing”:

“Similarly, further classification (using masks 2 and 3) was performed in states III and IV to reveal eL40 (State III, subclass 1) and the retracted L1-stalk (State V, subclass 1), respectively.”

4) Within the pdbs of states I-VI zinc ions are missing in several ribosomal proteins (including eL40) as well as the two zinc ions shown in Figure 2A, which belong to Nmd3. Given the high quality of these reconstructions and their ability to serve as future structural references, adding zinc ions would be very helpful for the scientific community.

Thank you for pointing out this omission. We have updated the pdb files.

5) Since there are six different states, it would be helpful to indicate in each figure, from which state a particular view originates. With this information the reader will be able to look at the correct pdb and EM map at the same time.

Thank you for the suggestion. Done.

6) Local resolution is an important factor to consider for the interpretation of all assembly factors involved in this study. While Figure 1—figure supplement 1F shows a distribution of resolutions, it would be helpful to have a supplementary figure to Figure 1A to show local resolutions for all discussed ribosome assembly factors. These resolution estimations could be visualized using the latest features of Relion3 where low-pass filtering is also applied to these regions at their respective local resolution. Along similar lines, figures that show side chain and bases should be backed up with corresponding supplementary figures showing the corresponding density – ideally not low-pass filtered.

We have added Figure 1—figure supplement 4 to address this point. We also show density for the side-chain base interactions in Figure 2B, C and Figure 6 A-C.

Reviewer #2:…Specific comments regarding the presentation of the data:– Abstract: The Abstract should be designed in the following way: introduction, main results, conclusion, importance. The first sentence is related to the importance and should be placed accordingly. In the second sentence, 'entire dynamic landscape' should be removed, because authors don't really know the entire landscape. It is also certainly not 'dynamic' because only individual snapshots are presented by cry-EM. Last sentence, the data concerns yeast ribosome, therefore leukemia related effects can only be suggested, whereas if authors are interested in 'illuminating', they should have worked with the relevant mutants of human ribosomes.

We have amended the title and Abstract as suggested.

Title: “Mechanism of completion of peptidyltransferase centre assembly in eukaryotes”

Abstract:

“During their final maturation in the cytoplasm, pre-60S ribosomal particles are converted to translation-competent large ribosomal subunits. Here, we present the mechanism of peptidyltransferase centre (PTC) completion that explains how integration of the last ribosomal proteins is coupled to release of the nuclear export adaptor Nmd3. Single-particle cryo-EM reveals that eL40 recruitment stabilizes helix 89 to form the uL16 binding site. The loading of uL16 unhooks helix 38 from Nmd3 to adopt its mature conformation. In turn, partial retraction of the L1 stalk is coupled to a conformational switch in Nmd3 that allows the uL16 P-site loop to fully accommodate into the PTC where it competes with Nmd3 for an overlapping binding site (base A2971). Our data reveal how the central functional site of the ribosome is sculpted and suggest how the formation of translation-competent 60S subunits is disrupted in leukaemia-associated ribosomopathies”

– Results and Figures:While six high resolution maps were obtained in this study (states I-VI), only two models were built (states I and III), and some of the comparison is done on the level of the density maps. I consider it as a problem because it limits readers' accessibility to the reported information. There are certain basic standards in structural biology implying that the experimental data (density maps) are interpreted into models (atomic coordinates), which can be validated according to the tools originally developed for X-ray crystallography, the validated models are then used to discuss the molecular mechanisms. Here authors compromise those standards by skipping validation of the model building for states II, IV, V, VI or at least it is not reported in the current version of the manuscript. The authors should build the models, validate all of them, upload all the structural information to the public domain and present the validation details in the supplementary materials. Currently, Supplementary file 1A shows data from two models, whereas Supplementary file 1B shows data for a single model, the rest is missing.

Apologies for the confusion, but in fact the models for all six states were indeed built and submitted together with the validation reports and the maps for the initial review process and were commented on by reviewer 1. We have deposited the maps and models for states I-VI in the public domain and present the data collection, model validation and refinements in Supplementary file 1A and B.

All the figures should have consistent representation of the densities and the models. For example in Figure 2C and 2D the density is shown in two different modes and not in the same colors.

Apologies, we have corrected this.

Figure 4A and 4D describes flexibility if H89. How can an atomic model from one study be compared with experimental density from another study? Please compare density with density or model with model.

We apologise that our labelling of Figure 4 caused confusion. We were comparing density with an atomic model from our own study. We have amended Figure 4A to clarify that we are comparing the densities for H89 that we observed in our focused classification of state I. The atomic model for the tip of H89 that we have built in state IV is now shown separately in Figure 4B.

Also for Figure 5; in 5A-C, it is the overall models that should be compared, especially given that the authors claim near atomic structures; in the bottom insets, the density should be shown in blue mesh, which is the standard for high resolution work and provides evidence of experimental data. The same goes for 5E, the current representation is not possible to track.

We have modified the figures to take account of the reviewer’s comments, showing mesh in Figures 5A-C. Figure 5D shows atomic models of the closed and partially retracted states of Nmd3 for clarity.

– Discussion:There is quite a gap between reporting a structure of yeast ribosome to discussing a mechanism related to inherited forms of leukemia. The association of the clinical phenotype is certainly important, but such an implication should be justified and conservation between yeast and human ribosomes should be illustrated in the supplementary material.

As the reviewer points out, our conclusions regarding the mechanism of PTC assembly are likely to be common to all eukaryotes. To support the discussion on the clinical relevance of our findings, we have added Figure 7—figure supplement 1 to show the conservation of uL16 sequence and structure. We have also included the following text:

“Importantly, given the high degree of conservation in the amino acid sequence and structure of the uL16 protein (Figure 7—figure supplement 1A, B)”